# Prenatal exposure to Zika virus shapes offspring neutrophil function in a sex-specific manner

Jiahui Ding [1,2] ✉, Anna Hu[1,3], Annie Thy Nguyen [1,4], Grace M. Swanson [1,5], Aditi Singh[1], Nicholas Adzibolosu [1], Diana Manchorova[1], Elizabeth Findeis[1], Anthony Maxwell[1], Yuan He[2], Marta Rodriguez Garcia[1,2] & Gil Mor[1]

Maternal viral infection during pregnancy can have lasting consequences on offspring immune development. Zika virus (ZIKV) is known to trigger maternal immune activation (MIA), yet its impact on fetal and postnatal innate immunity remains poorly understood. Here, we investigate how prenatal exposure to ZIKV influences offspring neutrophil function using a murine model of maternal infection. We identify a sex-dimorphic placental response to ZIKV and observed hyperinflammation in ZIKV-exposed male offspring following LPS challenge. Functional assays reveal impaired reactive oxygen species production and defective neutrophil extracellular trap formation in neutrophils from ZIKV-exposed offspring. Furthermore, we identify A20 as a key sex-dimorphic regulator of neutrophil activation and survival. Here, we show that maternal viral infection during pregnancy programs long-term offspring immunity in a sex-specific manner, providing insights into the developmental origins of differential susceptibility to infections and inflammatory diseases later in life.

The Developmental Origins of Health and Disease (DOHaD) hypothesis[1] suggests that fetal immune system development begins *in utero* and is shaped by maternal immune status during pregnancy. Maternal infection may uniquely alter the molecular and cellular elements at the maternal-fetal interface and reprogram the fetal immune system, with effects that extend beyond birth and alter their immune responses to vaccines and infections later in life[2]. However, the impact of maternal viral infection on offspring long-term immunity has not yet been fully defined. As we confront a growing risks of pandemics, such as SARS (2002), Ebola (2013), Zika (2016), and COVID-19 (2019), billions of children have been born under the shadow of maternal viral infections, emphasizing the urgent need to understand the effects of maternal viral infection on long-term offspring immunity and the underlying mechanisms.

Zika virus (ZIKV), a positive-sense, single-stranded RNA virus of the *Flavivirus* genus[3], is primarily transmitted to humans through the bites of *Aedes* mosquitoes but can also spread via sexual transmission and vertical transmission from mother to fetus[4]. As one of the newly defined "TORCH" pathogen[5], maternal ZIKV infection leads to adverse pregnancy outcomes, including preterm birth, stillbirth and congenital Zika syndrome (CZS). Infants with CZS may show microcephaly, abnormal brain development, limb contractures, eye abnormalities and other neurologic manifestations[5,6]. While most research has focused on ZIKV's impact on fetal/offspring brain development[7,8], ZIKV exposures *in utero* also have long-term consequences on the immune system of the offspring, altering their immune responses to childhood vaccines and potentially increasing their susceptibility to future infections[9]. Although the 2016 ZIKV outbreak has declined, ZIKV

[1]C.S Mott center for Human Growth and Development, Department of Obstetrics and Gynecology, Wayne State University, Detroit, MI, USA. [2]Department of Biochemistry, Microbiology and Immunology, Wayne State University, Detroit, MI, USA. [3]Center for Vaccines and Immunology, Department of Infectious Disease, University of Georgia, Athens, GA, USA. [4]School of Medicine, Wayne State University, Detroit, MI, USA. [5]Genome Sciences Core, Wayne State University, Detroit, MI, USA. ✉e-mail: candy.ding@wayne.edu

remains endemic in many regions[10], with re-emergence risk amplified by global warming and vector persistence[11].

ZIKV infection during pregnancy triggers maternal immune activation (MIA) at the maternal-fetal interface, with placental immune and trophoblast cells producing a range of proinflammatory cytokines and chemokines, such as IL-1β, IL-6, TNF-α, and type I interferons[6,12]. These inflammatory mediators, while critical for controlling infection, may also affect the development of fetal immune cells, influencing long-term immune responses in the offspring. Interestingly, our recent findings demonstrate significant sex-dimorphic differences in female and male placental responses to identical stimuli within the same uterus[13], suggesting that maternal ZIKV infection may program distinct immune trajectories in male and female offspring.

Among the immune cells that may be altered by MIA, neutrophils are particularly important due to their role in the innate immune response. Neutrophils are the most abundant white blood cells in humans and are a critical first responder cells of the innate immune system against infections[14]. Dysregulated neutrophil immunity contributes to severe infection, massive tissue injury and failure of inflammation resolution[15]. Children aged 9–65 months with CZS and microcephaly have a significant decrease in the number of normal neutrophils and a dramatic increase in the number of hyper-segmented and vacuolated neutrophils[16], highlighting neutrophil abnormalities in ZIKV-exposed children with microcephaly. However, neutrophil dysfunction may also occur in ZIKV-exposed infants without congenital abnormalities observed. Among women with confirmed or possible Zika infection during pregnancy in U.S. states and territories, Zika-associated birth defects occurred in about 5% of babies[17]. The remaining 95% of babies may appear clinically normal at birth but could exhibit abnormal immune responses when challenged by future infections. Thus, long-term follow-up of ZIKV-exposed infants including immune function assessment is essential. To date, few studies have investigated the long-term immune consequences of maternal ZIKV infection, particularly regarding neutrophil function in the offspring. This represents a critical knowledge gap, as maternal ZIKV infection may alter offspring neutrophil function, potentially impairing immunity against subsequent bacterial and viral challenges.

In this study, we investigate the long-term effects of maternal ZIKV infection on offspring immunity using a mouse model. We identify a sex-dimorphic placental response to ZIKV and observed hyperinflammation in ZIKV-exposed male offspring following LPS challenge. Functional assays reveal compromised reactive oxygen species (ROS) production and impaired neutrophil extracellular trap (NET) formation in neutrophils from ZIKV-exposed offspring. Furthermore, we identify A20 as a key sex-dimorphic regulator of neutrophil activation and survival. Our findings provide insights into the long-lasting sex-specific impact of maternal viral infection on offspring immunity, advancing our understanding of how maternal immune activation shapes long-term offspring immune function.

## Results

### Impact of Zika viral infection during pregnancy on offspring growth

To investigate the long-term impact of mild ZIKV infection during pregnancy on offspring growth, we administered a low dose of Zika virus (50 PFU/mouse) via intraperitoneal injection to pregnant mice at embryonic day 8.5 (E8.5) and assessed fetal and postnatal growth. Pregnancy outcomes, including implantation number, fetal resorptions, gestational length, litter size, sex ratio and placental weight, showed no significant differences between control and Zika-exposed groups at E12.5 or E17.5 (Fig. 1a–d). At E12.5, placental and fetal weights did not differ significantly between the control and Zika-exposed groups, irrespective of fetal sex (Fig. 1e). By E17.5, ZIKV-exposed male placentas weighed significantly more than female placentas (Fig. 1f). In

terms of fetal weight, control male fetuses were heavier than control female fetuses. Notably, Zika-infection had an effect only in the development of male fetuses, exposed male fetal weight was significantly reduced compared to control male fetal weight, whereas female fetal weights remained unaffected by Zika exposure (Fig. 1f). This male-biased effect is consistent with clinical observations that male sex is significantly associated with abnormal developmental outcomes in children prenatally exposed to ZIKV[18]. To evaluate placental efficiency, we calculated the fetal-to-placental weight ratio, no significant differences were observed between control and ZIKV-exposed groups in either male-or female placentas at E12.5 or E17.5 (Supplementary Fig. 1). To further evaluate fetal growth, we measured the crown-rump length (CRL) and occipital-frontal diameter (OFD) (Fig. 1g). At E12.5, no differences were observed in CRL or OFD between groups (Fig. 1h, i). By E17.5, CRL measurements revealed a consistent sex associated developmental pattern: control male fetuses had significantly longer CRL than control females, and Zika-exposed male CRL was significantly reduced compared to control males (Fig. 1h). OFD measurements, however, did not show any differences across groups or sexes at either E12.5 or E17.5 (Fig. 1i).

To determine whether the sex-specific developmental differences observed *in utero* persist postnatally, we measured body weight at postnatal days 35–38 (PND35-38). Control males weighed significantly more than control females; however, Zika-exposed male offspring showed a significant reduction in body weight compared to control males, whereas female offspring were unaffected by Zika exposure (Fig. 1j). These findings highlight that Zika exposure selectively impairs male growth, with effects extending from fetal to postnatal development.

### Immune response to Zika viral infection in the placenta

Our next objective was to elucidate the immunological pathways activated during pregnancy that could impact fetal male development. In immunocompetent mice, Zika viral infection is constrained due to a potent Type I interferon response abrogating virus replication[19,20]. To investigate whether there is a sex-dimorphic response to a low dose of maternal Zika virus infection in the placenta, we administered a low dose of Zika virus (50 PFU/mouse) via intraperitoneal injection to pregnant mice at E8.5 to assess immune activation in whole placental tissues by flow cytometry and bulk RNA sequencing. Firstly, no detectable viral titers were observed in the placentas at E12.5 or E17.5 (4 and 9 days post infection, respectively) (Supplementary Table 1), confirming the capacity of the placenta to control the infection and prevent ZIKV vertical transmission, consistent with previous studies using much higher doses of ZIKV exposure[21]. Next, maternal serum cytokine levels at E12.5 showed no significant differences between control and ZIKV-infected dams (Supplementary Table 2), suggesting that maternal systemic inflammation was well controlled. Given the absence of systemic inflammation, we next assessed whether local inflammatory responses were present in the placenta. In the placenta at E12.5, Zika virus exposure led to a significant increase in myeloid cells (CD45+CD11b+) in both male and female placentas (Fig. 2a, c). By E17.5, however, myeloid cell levels had decreased in male placentas and were no longer significantly different from controls, whereas in female placentas, the increase in CD45+CD11b+ myeloid cells persisted (Fig. 2b, d), revealing a dynamic and sex-specific placental immune profile following maternal ZIKV infection.

Given the robust and sex-specific placental immune response observed at E12.5, we aimed to better understand the signals involved in these responses. Thus, we performed bulk RNA sequencing in normal and ZIKV exposed placentas ($n = 3-5$ female or male placentas from 2 dams per group; placentas were selected from both uterine horns). Principal component analysis (PCA) of the placental transcriptome revealed high variability, likely attributable to litter effects and biological heterogeneity. Previous observations in rats[22] and

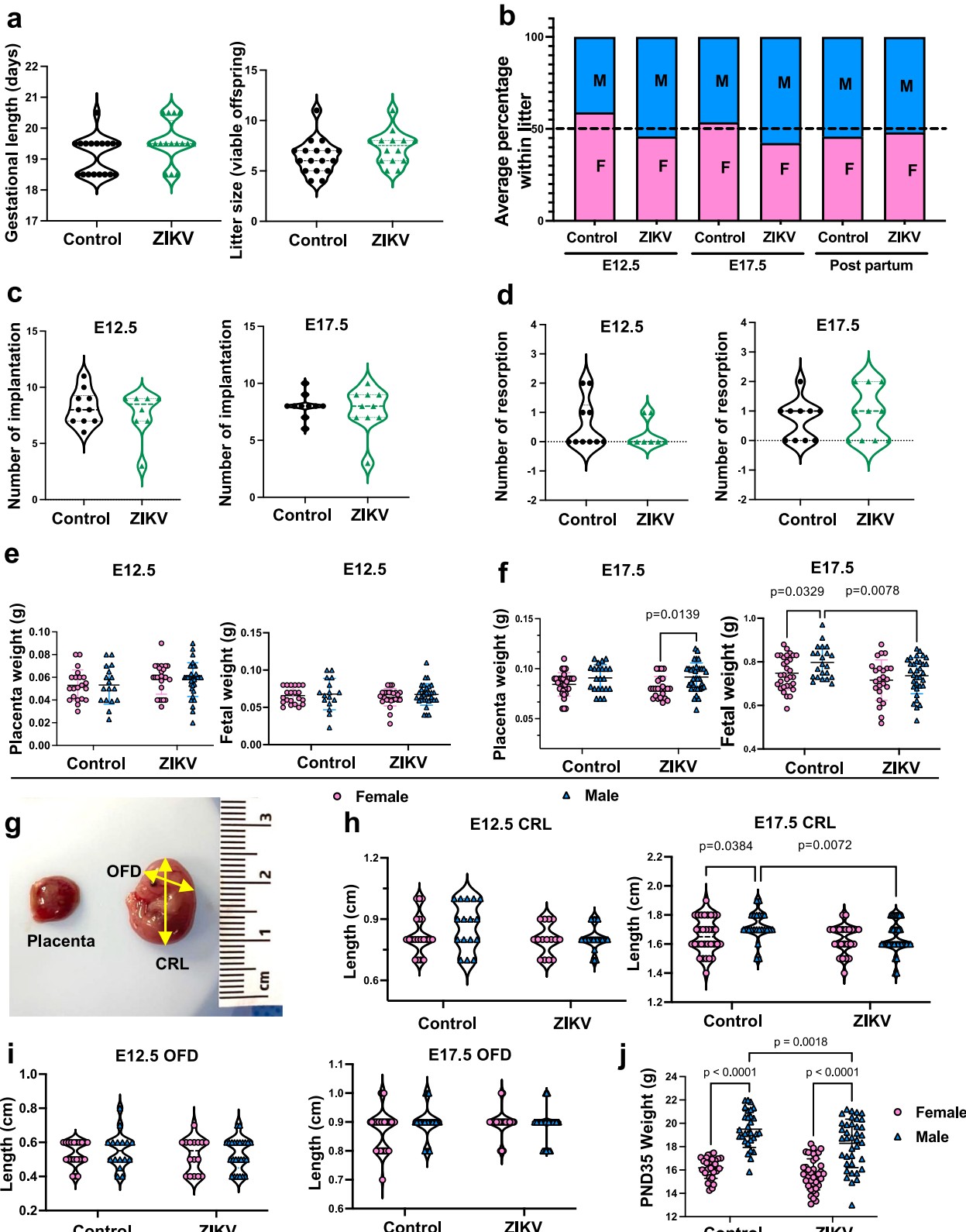

recent studies in mice[23] indicate that fetuses and placentas are not equally affected in these multiple-litter-bearing species, displaying intra-litter variability influenced by factors such as horn side and intrauterine position. Therefore, future RNA-seq studies focusing on placental sex dimorphism in mice and rats should adopt more rigorous standards for sample selection, including consideration of litter size, uterine horn side, and intrauterine position.

As shown in Fig. 3a, PCA showed better separation in male placentas compared to females, suggesting potential sex-specific transcriptional differences. Analysis of differentially expressed genes (DEGs) between control and zika-exposed female and male placentas highlighted unique sex-dimorphic transcriptome patterns in response to Zika exposure in the placentas (control female placenta vs. Zika-exposed female placenta; control male placenta vs. Zika-exposed male

**Fig. 1 | The impact of maternal Zika virus infection on pregnancy outcomes and offspring growth. a** Gestational length (*n* = 18 control, 16 ZIKV) and litter size (*n* = 17 control, 14 ZIKV). **b** Average percentage of female and male fetuses at E12.5, E17.5, and postnatally. **c**, **d** Implantations and resorptions at E12.5 (*n* = 10 control, 8 ZIKV) and E17.5 (*n* = 10/group). **e** Placenta weight and fetal weight at E12.5 by sex (*n* = 21 female, 17 male placentas/fetuses from 5 control dams; *n* = 25 female, 33 male placentas/fetuses from 7 ZIKV dams). **f** Placenta weight and fetal weight at E17.5 by sex (*n* = 32 female, 22 male placentas/fetuses from 7 control dams; *n* = 24 female, 34 male placentas/fetuses from 7 ZIKV dams). **g** Representative placenta and fetus image. **h** CRL at E12.5 (*n* = 20 female, 16 male fetuses from 5 control dams;

*n* = 15 female, 25 male fetuses from 7 ZIKV dams) and E17.5 (*n* = 32 female, 23 male fetuses from 7 control dams; *n* = 24 female, 34 male fetuses from 7 ZIKV dams) by sex. **i** OFD at E12.5 (*n* = 21 female, 16 male fetuses from 5 control dams; *n* = 16 female, 25 male fetuses from 7 ZIKV dams) and E17.5 (*n* = 32 female, 23 male fetuses from 7 control dams; *n* = 24 female, 34 male fetuses from 7 ZIKV dams) by sex. **j** Postnatal body weight at day 35 (*n* = 29 female, 30 male offspring from 7 control dams; *n* = 39 female, 38 male offspring from 10 ZIKV dams) by sex. Data are mean ± SD. Two-sided Student's *t*-test (**a**, **c**, **d**) and two-way ANOVA with Šídák's multiple comparisons (**e**, **f**, **h**, **i**, **j**). M male, F female, OFD occipital–frontal diameter, CRL crown–rump length. Source data provided.

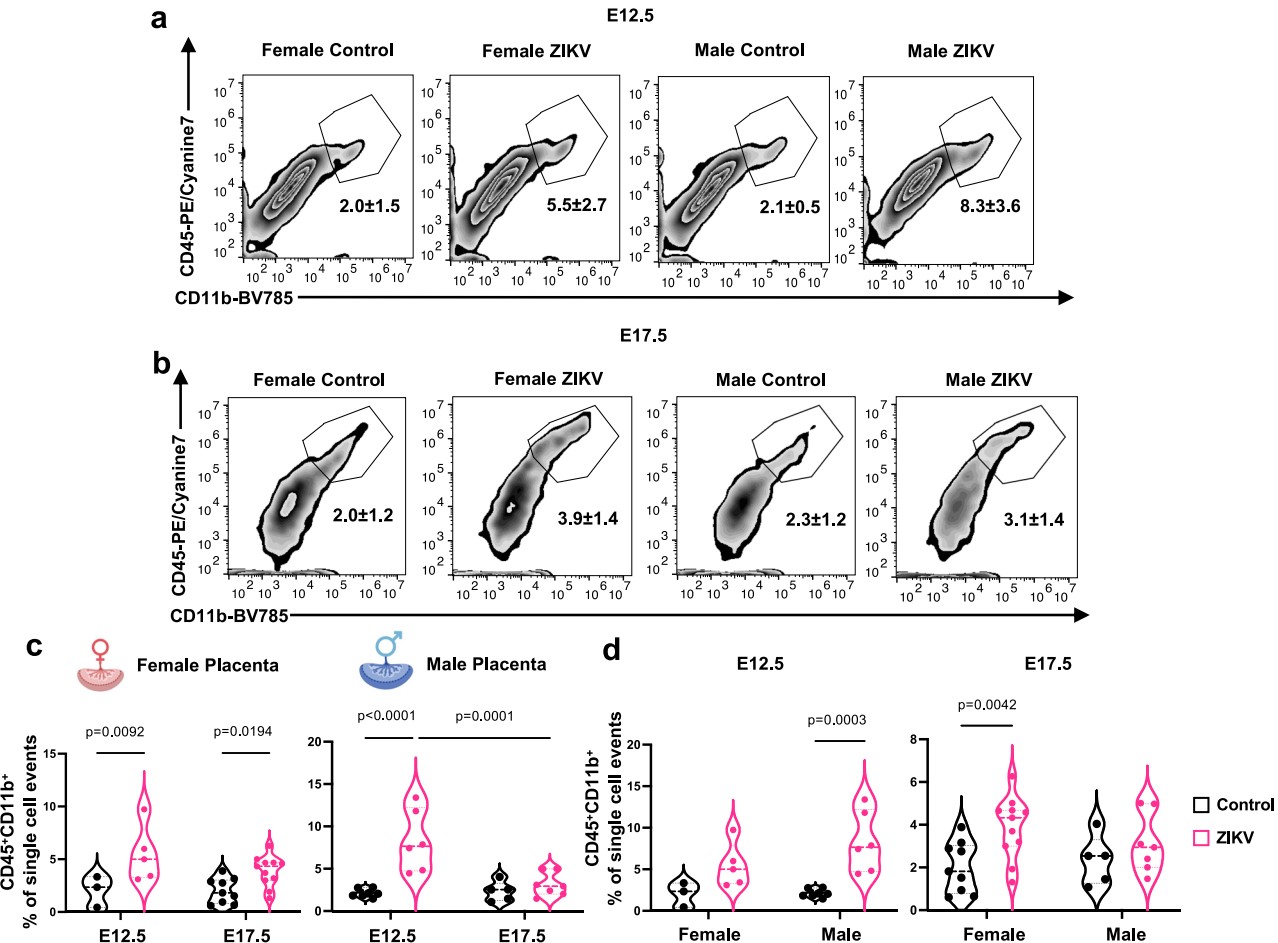

**Fig. 2 | ZIKV infection increases CD45⁺CD11b⁺ myeloid cells in the placenta in a sex-specific manner.** At E8.5, pregnant mice were intraperitoneally injected with ZIKV (50 PFU/mouse), and myeloid cells population in the placenta was evaluated by flow cytometry at E12.5 and E17.5. **a**, **b** Representative scatter plots of CD45⁺CD11b⁺ populations in the placenta at E12.5 and E17.5. **c** Percentage of CD45⁺CD11b⁺ populations in female and male placentas at E12.5 (*n* = 3 female, *n* = 7 male placentas from 2 control dams; *n* = 5 female, *n* = 6 male placentas from 2 ZIKV

dams) and E17.5 (*n* = 9 female, *n* = 5 male placentas from 4 control dams; *n* = 11 female, *n* = 7 male placentas from 4 ZIKV dams). **d** Re-analysis of placental CD45⁺CD11b⁺ populations at E12.5 and E17.5 by sex. Two-way ANOVA with Šídák's multiple comparisons was used. Source data are provided as a Source Data file. Female and male placenta symbols (panels **c** and **d**) were created in BioRender. Alvero, A. (2025) http://BioRender.com/x5k7dzh.

placenta) (Fig. 3b). The DEGs identified between female and male control placentas, as well as between female and male ZIKV-exposed placentas, indicate baseline sex dimorphism in the placental transcriptome and sex-specific transcriptional responses to viral exposure (Supplementary Fig. 2). To identify shared and sex-specific biological processes in placental responses to ZIKV, we performed a meta-analysis with high-specificity pruning. Only cholesterol homeostasis was common to both sexes, while 25 processes were unique to male placentas and 4 processes to female placentas (Fig. 3c). Male placentas showed enrichment of innate immune pathways, including interferon-beta (IFN-β), signaling, interleukin-1 beta (IL-1β) regulation, and

neutrophil activity, whereas female placentas were enriched for tissue organization and metabolic processes (Fig. 3c). Pathway analysis further corroborated these findings, with male placentas uniquely activating immune-related pathways such as NOD-like receptor, HIF-1 signaling, and NET formation. In contrast, female placentas exhibited activation of metabolic pathways, including glutathione metabolism, histidine metabolism, and mineral absorption (Fig. 3d). These results highlight distinct sex-specific biological and molecular responses to ZIKV in the placentas.

To validate these transcriptomic findings, we measured mRNA expression of key immune genes using qRT-PCR in male and female

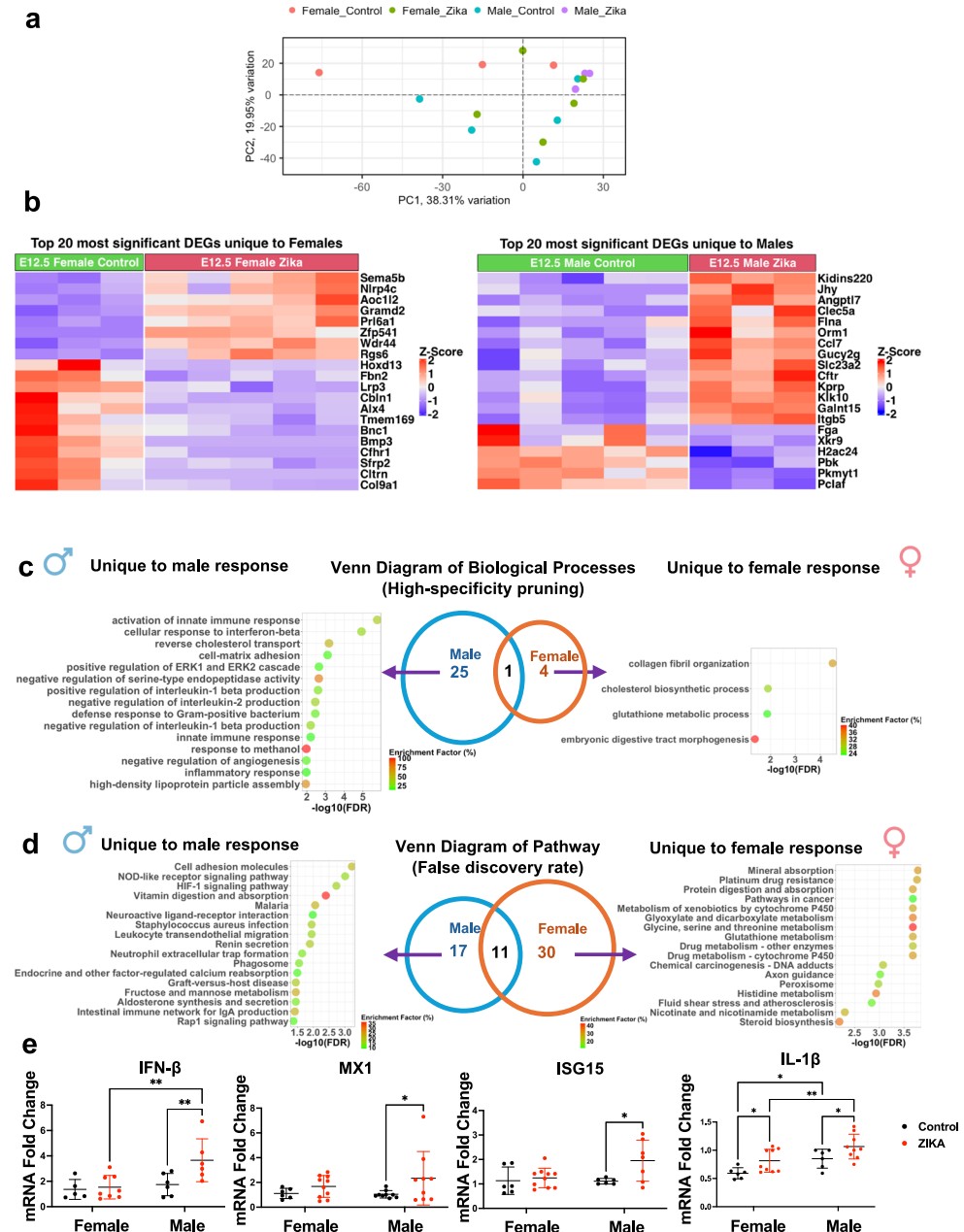

**Fig. 3 | Zika infection induces a sex-dimorphic response in the mouse placenta.**
At E8.5, pregnant mice were intraperitoneally injected with ZIKV (50 PFU/mouse), and placentas were collected for bulk RNA sequencing at E12.5. **a** PCA plot of female control placenta (n = 3), female ZIKV placenta (n = 5), male control placenta (n = 5), and male ZIKV placenta (n = 3), placentas were collected from 2-3 dams per group. **b** Heatmap showing top 20 most significant differentially expressed genes (DEGs) (adjusted p value < 0.05) in female and male responses. **c** Venn diagram of top biological processes unique to females and males' response to ZIKV after high-specificity pruning. **d** Venn diagram of top pathways unique to females and males' response to ZIKV after false discovery rate correction. **e** mRNA expression of IFN-β (n = 5 female, n = 6 male placentas from 3 control dams; n = 8 female, n = 6 male placentas from 3 ZIKV dams; p = 0.0077 male control vs. male ZIKV, p = 0.0022 female ZIKV vs. male ZIKV), Mx1 (n = 6 female, n = 9 male placentas from 3 control dams; n = 10 female, n = 9 male placentas from 3 ZIKV dams; p = 0.0342 male control vs. male ZIKV), Isg15 (n = 6 female, n = 7 male placentas from 3 control dams; n = 10 female, n = 7 male placentas from 3 ZIKV dams; p = 0.0455 male control vs. male ZIKV), and IL-1β (n = 6 female, n = 6 male placentas from 3 control dams; n = 9 female, n = 9 male placentas from 3 ZIKV dams; p = 0.033 female control vs. female ZIKV, p = 0.038 male control vs. male ZIKV, p = 0.0238 female control vs. male control, p = 0.0081 female ZIKV vs. male ZIKV) in E12.5 mouse placentas by qRT-PCR. Data are shown as the mean ± SD. Two-way ANOVA with Šídák's multiple comparisons was used. Source data are provided as a Source Data file.

placentas exposed to Zika infection. Male placentas showed significant upregulation of IFN-β, interferon-stimulated genes (ISGs) such as Mx1 and ISG15, and IL-1β. While IL-1β expression was also significantly elevated in female placentas, the increase was less pronounced compared to males (Fig. 3e). These data demonstrate sex-dimorphic placental responses, with male placentas exhibiting a stronger activation of immune-related signaling pathways, including IFN-β

and IL-1β responses, while female placentas showing mainly metabolic adaptations.

## Effect of maternal Zika infection on offspring immune response to LPS

To evaluate the long-term consequences of maternal ZIKV infection on offspring immunity, male and female offspring from control and Zika-

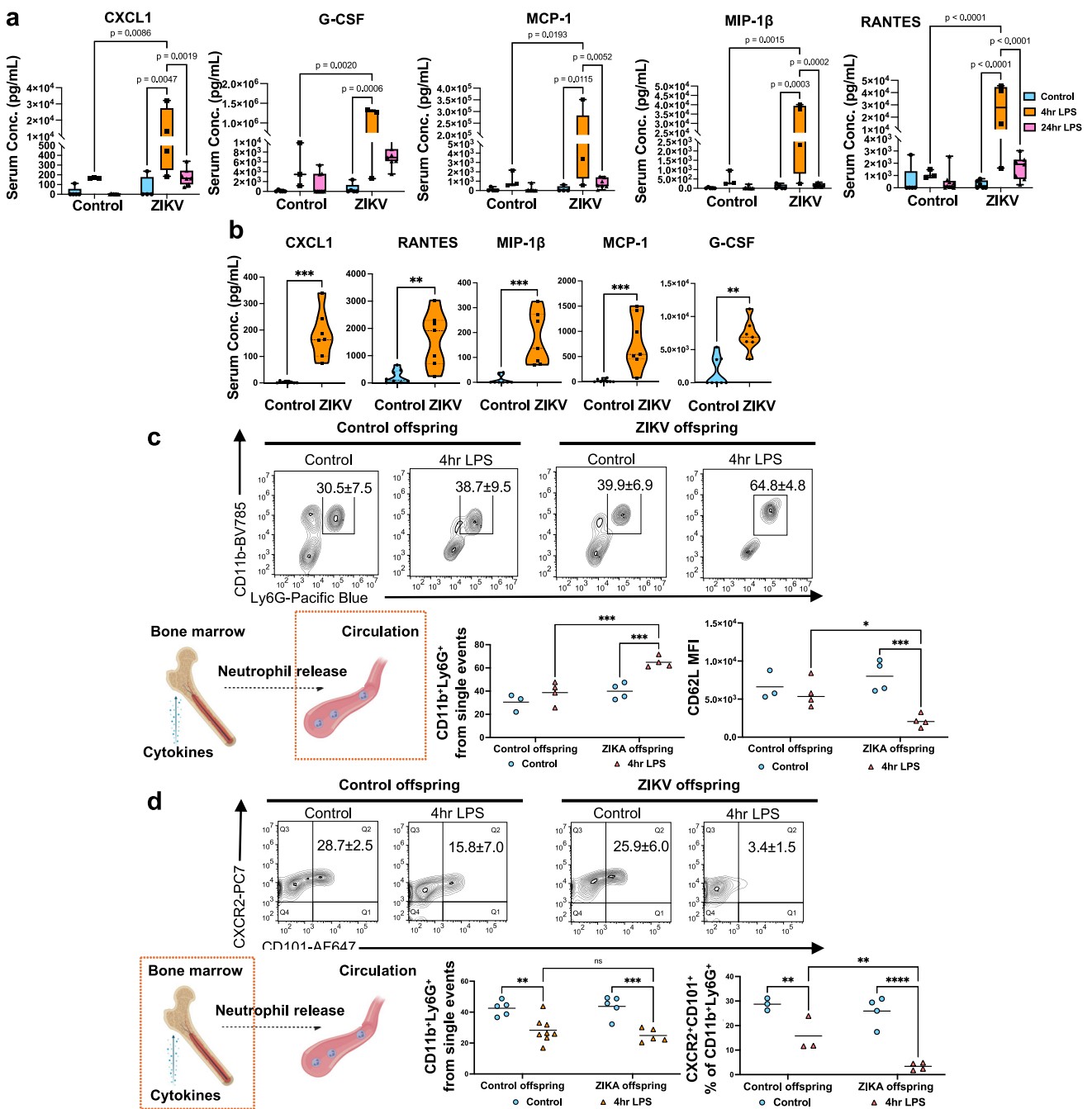

**Fig. 4 | Effect of maternal Zika infection on offspring immune response to LPS.**
**a** Serum CXCL1, G-CSF, MCP-1, MIP-1β, and RANTES at 4 and 24 h post-LPS in male offspring. Controls: untreated ($n = 5$), 4 h LPS ($n = 3$), 24 h LPS ($n = 9$). ZIKV: untreated ($n = 6$), 4 h LPS ($n = 4$), 24 h LPS ($n = 7$). **b** Serum cytokines at 24 h LPS: control ($n = 8$) vs. ZIKV ($n = 7$); CXCL1 ($p = 0.0003$), RANTES ($p = 0.0022$), MIP-1β ($p = 0.0003$), MCP-1 ($p = 0.0006$), G-CSF ($p = 0.0012$). **c** Neutrophil release from bone marrow (Created in BioRender. Alvero, A. (2025) http://BioRender.com/lgg2lp5). Blood CD11b⁺Ly6G⁺ neutrophils and CD62L mean fluorescence intensity (MFI) at 4 h LPS: control untreated ($n = 3$), control 4 h LPS ($n = 4$), ZIKV untreated ($n = 4$), ZIKV 4 h LPS ($n = 4$). CD11b⁺Ly6G⁺: ZIKV control vs. 4 h LPS ($p = 0.0005$),

control vs. ZIKV 4 h LPS ($p = 0.0004$). CD62L MFI: ZIKV control vs. 4 h LPS ($p = 0.0005$), control vs. ZIKV 4 h LPS ($p = 0.0123$). **d** Bone marrow CD11b⁺Ly6G⁺ neutrophils (control untreated $n = 5$, control 4 h LPS $n = 8$, ZIKV untreated and 4 h LPS $n = 5$) and CXCR2⁺CD101⁺ mature subsets (control $n = 3$, ZIKV $n = 4$) at 4 h LPS. CD11b⁺Ly6G⁺: control untreated vs. 4 h LPS ($p = 0.0012$), ZIKV control vs. 4 h LPS ($p = 0.0002$). CXCR2⁺CD101⁺: control untreated vs. 4 h LPS ($p = 0.0078$), ZIKV control vs. 4 h LPS ($p < 0.0001$), control vs. ZIKV 4 h LPS ($p = 0.0068$). Data are mean ± SD. Two-sided Student's *t*-test (b) and two-way ANOVA with Šídák's multiple comparisons (**a**, **c**, **d**) were used. Source data provided.

exposed groups were challenged with lipopolysaccharide (LPS) (0.8 mg/kg, i.p.) at PND35, when immune competency is established in rodents[24]. Serum cytokines and chemokines were measured at 4 h and 24 h post treatment using a 23-plex immunoassay (Bio-Rad). Typically, the acute inflammatory response is a highly coordinated sequence of events that initiates within minutes, peaks within a few hours, and

resolves by 24 h[25]. In control female and male offspring, cytokine and chemokine levels increased at 4 h post LPS treatment and returned to baseline by 24 h (Fig. 4a and Supplementary Fig. 3b). Interestingly, female control mice mounted a significantly stronger acute response than males, with higher CXCL1, G-CSF, MIP-1β, and RANTES (Supplementary Fig. 3a), consistent with previous reports demonstrating

greater cytokine upregulation in female mice compared to males in response to LPS[26]. Notably, Zika-exposed male offspring exhibited an exacerbated inflammatory response to LPS at 4 h compared to control male offspring, characterized by significant elevated levels of CXCL1, G-CSF, RANTES, MCP-1 and MIP-1β, a pattern not observed in female offspring (Fig. 4a and Supplementary Fig. 3b). By 24 h, control male offspring successfully resolved inflammation, but Zika-exposed male offspring retained elevated cytokine levels (Fig. 4b), suggesting delayed resolution and a risk of chronic inflammation and tissue damage. This prolonged inflammatory response was not observed in Zika-exposed female offspring (Supplementary Fig. 3c), indicating a sex-specific long-term effect of maternal Zika infection on offspring inflammatory regulation.

To further investigate the mechanisms underlying the differential inflammatory response to LPS in Zika-exposed offspring, we focused on neutrophils, as the elevated cytokines and chemokines are key regulators of neutrophil recruitment and activation[27–31]. Analysis of circulating neutrophils in the bloodstream revealed that Zika-exposed males had increased mature neutrophils with significant CD62L shedding-indicative of activation[32] after LPS challenge (Fig. 4c). In contrast, female offspring showed no change in neutrophil numbers but displayed a more activated phenotype (Supplementary Fig. 4a).

During acute inflammation, the bone marrow rapidly produces and releases neutrophils into the bloodstream, facilitating a robust immune response and supporting the resolution of inflammation (Fig. 4d). We hypothesized that differences in circulating neutrophils might reflect altered bone marrow release. Indeed, Zika-exposed male offspring reveled significantly fewer mature neutrophils (CXCR2$^+$CD101$^+$CD11b$^+$Ly6G$^+$) in the bone marrow after 4 h of LPS treatment compared to control male offspring (Fig. 4d), which coincided with the significant increased neutrophils in the bloodstream specifically observed in Zika-exposed male offspring. However, in the female offspring, bone marrow neutrophil numbers were unchanged between groups (Supplementary Fig. 4b). These findings suggest that maternal Zika infection induces a sex-dimorphic effect on neutrophil dynamics, specifically altering neutrophil mobilization and activation in male offspring but not females. Alternatively, neutrophils in ZIKV-exposed male offspring are either impaired in their ability to migrate into tissues − leading to their accumulation in the circulation − or exhibit delayed release from the bone marrow. As a result, elevated neutrophil levels are still detectable in the bloodstream of males at 4 h post challenge, whereas in females, neutrophils may have already exited the bloodstream and migrated into tissues.

## Maternal Zika infection impacts offspring neutrophil transcriptome

To fully understand the impact of maternal Zika infection on offspring neutrophil reprogramming, we isolated bone marrow-derived neutrophils from male and female offspring of control and Zika-exposed groups at PND35 and assessed transcriptomic and functional changes (Fig. 5a). Remarkably, we observed the presence of a sex-dimorphic transcriptomic profile in neutrophils from normal control female and male mice (Fig. 5b), 2260 DEGs were identified, with 1435 genes upregulated and 825 genes downregulated in the male neutrophils compared to the female neutrophils. *Xist*, a long non-coding RNA (lncRNA) that regulates X-chromosome inactivation (XCI) in female mammals[33] is highly expressed in female neutrophils, suggestive of a role for XCI in neutrophil sex-specific differentiation and function. On the other hand, in male neutrophils, we observed *Eif2s3y* and *Ddx3y*, two Y chromosome related genes differentially expressed (Fig. 5b). *Eif2s3y* plays a role in the pluripotency of embryonic stem cells[34]. *Ddx3y* is a gene that codes for an RNA helicase that's involved in RNA synthesis and degradation[35]. It is associated with male infertility but also regulates IFNβ expression and the inflammatory response to LPS[36]. We then analyzed the top biological processes, molecular functions, and pathways associated with DEGs (Fig. 5c) in neutrophils from normal male and female mice. Notably, pathways related to immune responsiveness, including "response to stimulus," "immune system process," "regulation of communication," "signaling receptor binding," "cytokine receptor activity," "TNF signaling pathway," "PI3K-Akt signaling pathway," and "JAK-STAT signaling pathway," were significantly enriched in female neutrophils. This suggests that female neutrophils may be more primed for activation and respond more rapidly to stimuli compared to male neutrophils.

Next, we evaluated whether ZIKV exposure during pregnancy will impact neutrophils transcriptome in a sex-specific manner. Thus, we conducted a meta-analysis comparing DEGs between male and female offspring bone marrow neutrophils from Zika-exposed dams. Our analysis revealed 53 shared DEGs between sexes, alongside 573 unique DEGs in male neutrophils and 300 unique DEGs in female neutrophils (Fig. 5d). In male neutrophils top DEGs, the upregulation of *Acod1* and *Cpa3* suggests alterations in immunometabolism and granule-associated functions. Additionally, changes in *Nrp2* and *Maff* indicate potential modifications in neutrophil migration and oxidative stress regulation. In contrast, female neutrophils exhibited DEGs involved in immune response and signaling regulation, including *Sp140l1*, *CD244a*, and *Zbp1*, pointing to altered neutrophil activation and antiviral defense mechanisms. Furthermore, changes in *Jak3* and *Bcl3* suggest modulation of cytokine signaling and transcriptional regulation, potentially implicating the JAK-STAT and NF-κB pathways. Pathway analysis revealed distinct sex-specific enrichment patterns. While "Metabolic pathways" was the only pathway shared between sexes, female-specific pathways included "TNF signaling pathway," "Chemokine signaling pathway," and "Pathways in cancer." In males, pathways such as "Neutrophil extracellular trap formation" and "Oxidative phosphorylation" were significantly enriched (Fig. 6a). These findings highlight sex-dimorphic reprogramming effect of *in utero* Zika virus exposure on offspring neutrophils, with male neutrophils exhibiting alterations in metabolic and extracellular trap-related processes, and female neutrophils displaying changes in immune signaling and cytokine regulation.

Neutrophils release web-like structures called neutrophil extracellular traps (NETs) to capture and kill pathogens in a process known as NETosis, which is essential for the host innate immune response[37]. Our findings revealed significant dysregulation of NETosis-related pathways in male neutrophils following *in utero* Zika virus exposure. Specifically, several histone genes, including *H2ac11*, *H2az2*, *H2ax*, *H4c14*, and *H2ac14*, were significantly downregulated in male neutrophils (Fig. 6b). Histones are essential for chromatin decondensation, a key step in NET formation, and their inhibition suggests impaired chromatin remodeling and NET release. Additionally, effector genes critical for NETosis, such as *Ctsg*, *Mpo*, and *Elane*, were also significantly decreased in male offspring from ZIKV infected dams (Fig. 6b). These data suggest a potential impairment on NETosis in male neutrophils from ZIKV infected dams.

## Impact of maternal Zika infection on offspring NETosis

Since *Ctsg*, *Mpo*, and *Elane* encode proteases and enzymes that mediate chromatin breakdown and antimicrobial activity, we posit that their downregulation compromised NETosis in male neutrophils. First, we validated the mRNA expression of *Mpo* and *Elane* in neutrophils using qRT-PCR. In male Zika offspring, *Mpo* expression was significantly reduced ($P = 0.0266$), whereas no significant change was observed in females ($P = 0.2104$). For *Elane*, expression levels were decreased in both female ($P = 0.0508$) and male ($P = 0.1982$) neutrophils, though these reductions did not reach statistical significance (Fig. 6c). These findings suggest that maternal Zika virus infection differentially impacts key NETosis-related genes in offspring neutrophils, with sex-specific variations in the magnitude and significance of these effects.

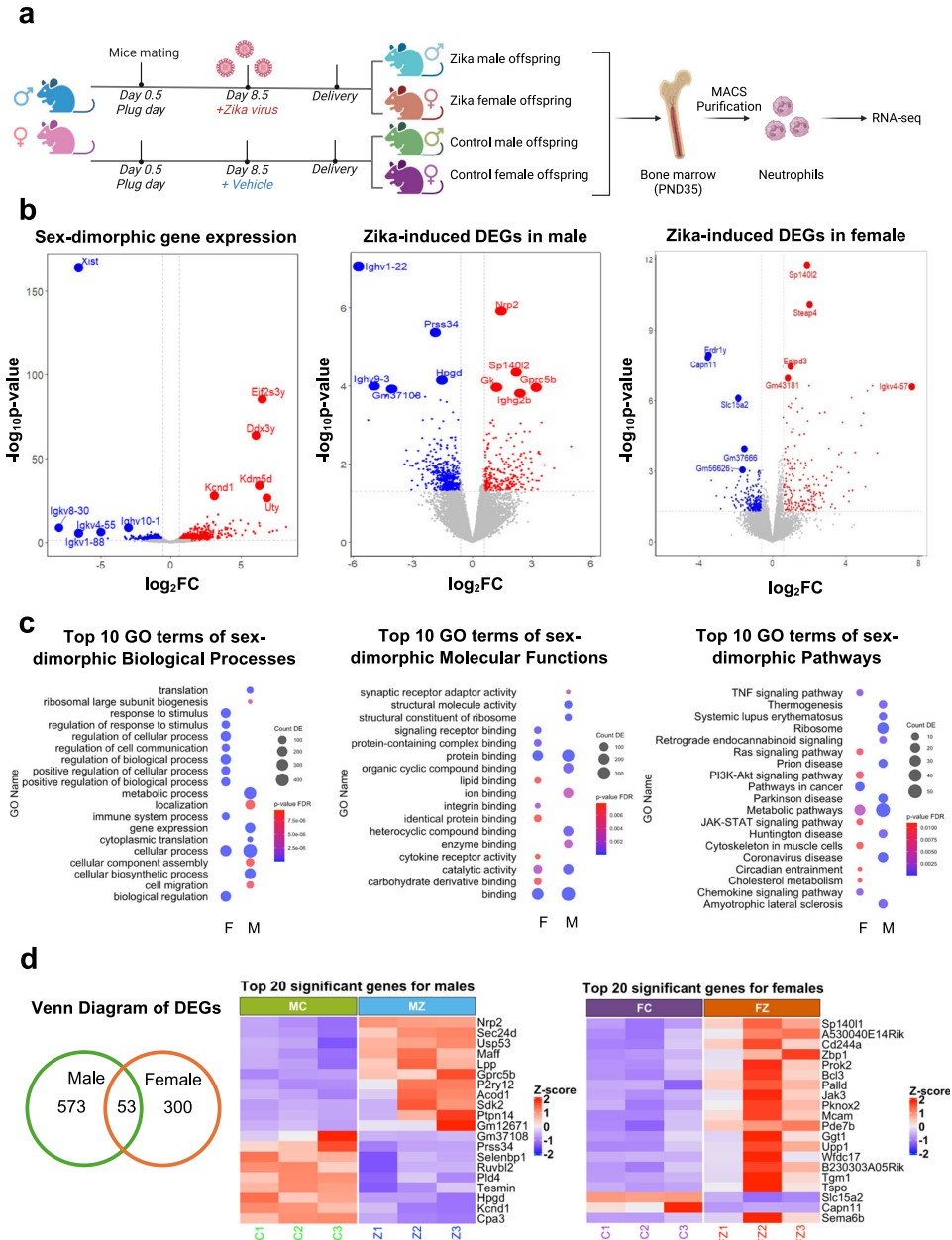

**Fig. 5 | Impact of maternal Zika infection on offspring neutrophil transcriptome. a** Experimental design. Bone marrow neutrophils were isolated from male and female offspring on PND35 from control and Zika-exposed groups for bulk RNA sequencing. Created in BioRender. Alvero, A. (2025) http://BioRender.com/nvwup3t. **b** Volcano plot of sex-dimorphic DEGs at baseline, DEGs in ZIKV-exposed male neutrophils, and DEGs in ZIKV-exposed female neutrophils. **c** Top 10 sex-dimorphic biological processes, molecular functions, and pathways comparing control female neutrophils and control male neutrophils. **d** Venn diagram of top DEGs unique to female and male offspring bone marrow neutrophils after *in utero* ZIKV exposure. Differential expression analysis was performed using DESeq2 with Benjamini–Hochberg correction for multiple comparisons. All tests were two-sided. The raw and processed RNA sequence data are available in NCBI's Gene Expression Omnibus under the accession number GSE292966.

Next, we assessed NETosis in offspring neutrophils following maternal ZIKV infection and their respective controls. Bone marrow-derived neutrophils from ZIKV-exposed and control offspring of both sexes were stimulated with 80 nM phorbol 12-myristate 13-acetate (PMA) to induce NET formation (Fig. 6d). NET formation was visualized by immunostaining using SYTOX Green, a DNA dye that labels extracellular DNA, and myeloperoxidase (MPO), a key NET-associated enzyme. Co-localization of SYTOX Green and MPO confirmed the presence of NETs; however, ZIKV-exposed offspring exhibited significantly reduced NET formation compared to controls, irrespective of sex (Fig. 6e and Supplementary Fig. 5). Since reactive oxygen species

(ROS) production is essential for NETosis[37], we next measured ROS production in the supernatant of isolated neutrophils from control and ZIKV-exposed offspring following PMA stimulation using the Cellular ROS Assay Kit (Abcam). Notably, ZIKV-exposed neutrophils from both sexes exhibited a significant reduction in ROS production compared to controls (Fig. 6f). These finding correlate with the transcriptomic analysis showing DEGs involved in ROS metabolism. Notably, Prdx4 and Mpo, both crucial for ROS production and scavenging, were downregulated in ZIKV-exposed neutrophils of both sexes (Fig. 6g). Collectively, these findings indicate that maternal ZIKV infection significantly impairs neutrophil function in offspring by reducing ROS

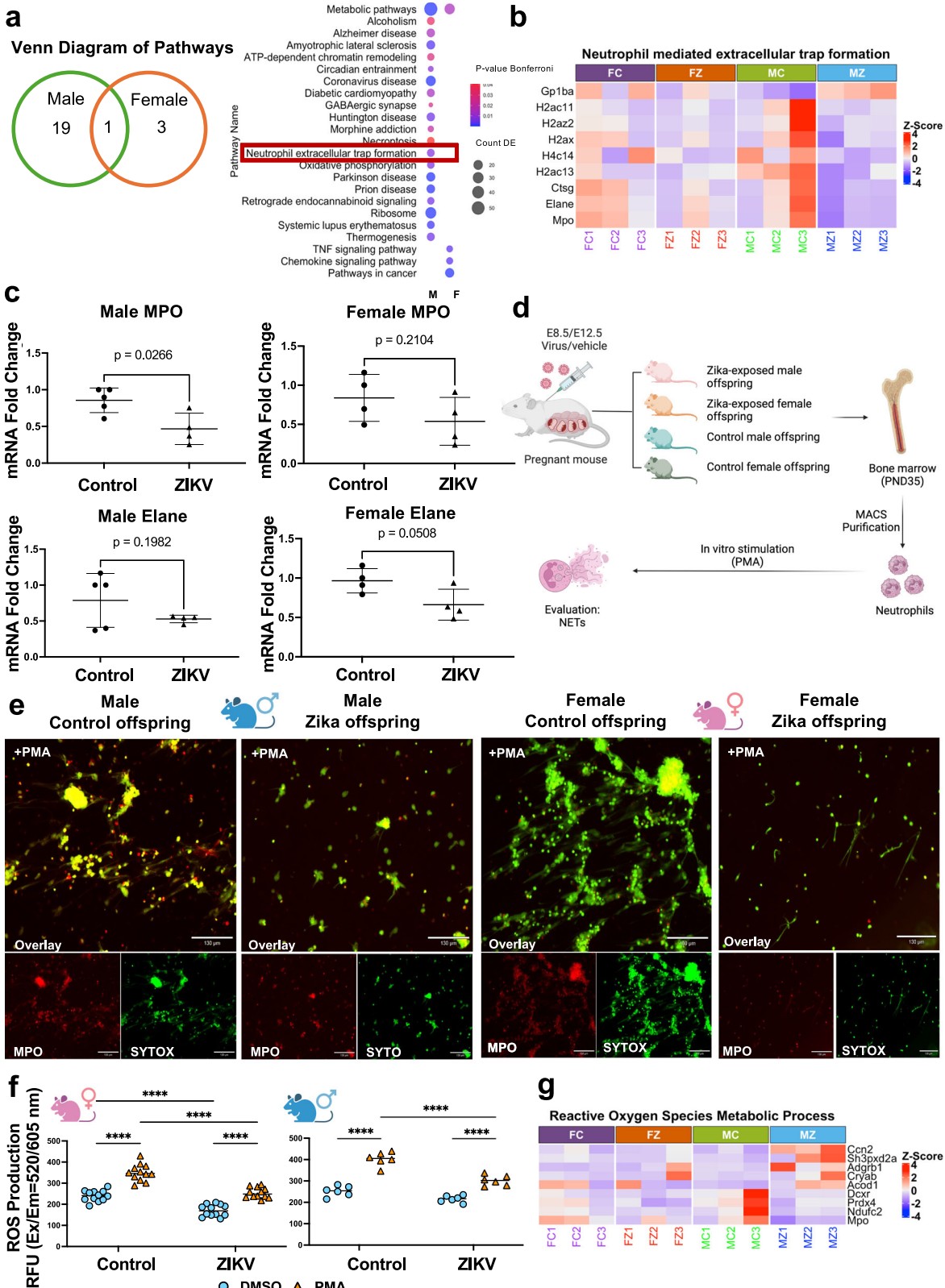

production and NET formation, potentially compromising offspring innate immune defenses against infections.

## Impact of maternal Zika infection on offspring neutrophil response

To further evaluate the impact of maternal Zika virus infection on offspring neutrophil function, we isolated and purified bone marrow neutrophils from ZIKV-exposed and control offspring of both sexes at PND35. Neutrophils were stimulated in vitro with 100 ng/mL LPS for 2 and 24 h to assess their activation status (Fig. 7a). In order to characterize the different stages of neutrophil differentiation and activation, we used CXCR2, a chemokine receptor critical for neutrophil migration and activation, that serves as a key marker of neutrophil functional states[38]. Within the isolated neutrophil populations, we

**Fig. 6 | Impact of maternal Zika infection on offspring NETosis. a** Venn diagram of top pathways unique to female and male offspring bone marrow neutrophils after *in utero* ZIKV exposure. Analysis was performed using DESeq2 with Benjamini–Hochberg correction for multiple comparisons. All tests were two-sided. **b** Heatmap of DEGs related to "Neutrophil-mediated extracellular trap formation." **c** mRNA expression of MPO and Elane in neutrophils from male and female offspring in control and ZIKV-exposed groups. For males: control (*n* = 5 isolations) and ZIKV (*n* = 4 isolations). For females: control (*n* = 4 isolations) and ZIKV (*n* = 4 isolations). Each isolation represents neutrophils pooled from the bone marrow of two mice, with each data point in the graph corresponding to one isolation. **d** Experimental design. Created in BioRender. Alvero, A. (2025) http://BioRender.com/bb2qwff. Bone marrow neutrophils were isolated from male and female offspring on PND35 from control and Zika-exposed groups and were treated with

phorbol 12-myristate 13-acetate (PMA) to evaluate NETosis. **e** Representative image of NET formation. Red: MPO; green: SYTOX, indicating extracellular DNA. Co-localization of SYTOX Green and MPO confirmed the presence of NETs. Scale bar = 130 μm. **f** ROS production of isolated neutrophils when treated with PMA for 45 min. Neutrophils were isolated from 2–4 mice per experiment across 3 independent experiments. ****$p$ < 0.0001. **g** Heatmap of DEGs related to "Reactive Oxygen Species Metabolic Process." Data are shown as the mean ± SD. Two-sided Student's *t*-test (**c**) and Two-way ANOVA with Šídák's multiple comparisons (**f**) were used. FC female control neutrophils, FZ female ZIKV neutrophils, MC male control neutrophils, MZ male ZIKV neutrophils, ROS reactive oxygen species. Female and male mice symbols (panel **e** and **f**) were created in BioRender. Alvero, A. (2025) http://BioRender.com/u0w40ys. Source data are provided as a Source Data file.

identified three distinct subsets based on CXCR2 expression levels: CXCR2$^{low}$, CXCR2$^{intermediate}$, and CXCR2$^{high}$.

Upon activation, neutrophils undergo phenotypic changes, including the shedding of CD62L (L-selectin) and upregulation of CD64 (FcγRI), which are well-established markers of neutrophil activation[39]. We further analyzed these markers to evaluate the activation status of the neutrophil subsets (Fig. 7b). Interestingly, our analysis revealed that CXCR2$^{intermediate}$ (CXCR2$^{int}$) and CXCR2$^{high}$ were the most responsive neutrophil subsets to LPS stimulation. A sex-dimorphic response to LPS was observed in neutrophils both at baseline and following *in utero* ZIKV exposure. In control offspring, after 2 h of LPS stimulation, male neutrophils exhibited a significantly higher percentage of activated cells within the CXCR2$^{int}$ subset compared to female neutrophils (Supplementary Fig. 6a). Although female neutrophils displayed a higher percentage of activated cells within the CXCR2$^{high}$ subset relative to males, there was no significant sex-specific difference in the LPS-induced activation within the CXCR2$^{high}$ subset (Supplementary Fig. 6a). At 24 h post LPS stimulation, the previously observed sex-dimorphic neutrophil response was no longer evident (Supplementary Fig. 6b), suggesting that the sex-specific response to LPS is time-dependent.

Furthermore, when comparing Zika-exposed offspring to their respective controls, we observed an altered neutrophil activation profile following LPS stimulation. In Zika-exposed male offspring, there was a significant increase in activated neutrophils within the CXCR2$^{int}$ subset compared to controls, and no significant change was observed in the CXCR2$^{high}$ subset (Fig. 7c). In contrast, neutrophils from female offspring exposed in uterus to ZIKV infection showed no difference in activation within the CXCR2$^{int}$ subset but revealed a significant reduction in activated neutrophils in the CXCR2$^{high}$ subset (Fig. 7d). By 24 h post LPS treatment, neutrophils from ZIKV-exposed male offspring, the increase in activated CXCR2$^{int}$ neutrophils was no longer apparent, but a significant reduction in activated neutrophils was observed in the CXCR2$^{high}$ subset compared to controls (Fig. 7e). Neutrophils from ZIKV-exposed female offspring, did not showed significant differences in activation across either subset (Fig. 7f). These findings reveal a dynamic, time-dependent, and sex-dimorphic modulation of neutrophil activation in response to LPS in both control and ZIKV-exposed offspring.

To further validate the differential stage of neutrophil activation, we collected conditioned media from in vitro LPS-treated neutrophils after 24 h of stimulation and screened for differences in the secretion of 23 inflammatory cytokines and chemokines by Luminex (Bio-rad). Neutrophils isolated from male and female control offspring exhibited a classical inflammatory cytokine response to LPS, but with distinct cytokine profiles. In male neutrophils, IL-6, TNF-α, G-CSF, and MIP-1α were the predominant cytokines induced by LPS, whereas in female neutrophils, the response was characterized primarily by IL-6, IL-1β, MIP-1β, and RANTES (Fig. 7g, h). Interestingly, neutrophils from ZIKV-exposed female and male offspring, although they showed an inflammatory response to LPS, the magnitude of this response was

significantly damped (Fig. 7g, h). These findings indicate that maternal ZIKV infection has a significant impact on the inflammatory response of offspring neutrophils to bacterial products, such as LPS.

## Role of A20 regulating neutrophil function

To elucidate the molecular mechanisms driving the dysregulated neutrophil function observed in Zika virus-exposed offspring, we aimed to identify key DEGs associated with the cellular response to LPS. Our analysis identified six common DEGs significantly linked to the LPS response in Zika-exposed neutrophils compared to controls (Fig. 8a). Among these, *Ly86*, *Ctsg*, and *Tnfaip3* (also known as A20) emerged as top candidates due to their critical roles in regulating inflammation. *Ly86* encodes a protein involved in TLR4 signaling[40], a key pathway in LPS-induced immune activation, while *Ctsg* (cathepsin G) is a serine protease that modulates inflammatory responses through the processing of cytokines and chemokines[41]. Notably, *Tnfaip3* (A20) is a central negative regulator of NF-κB signaling[42], a pivotal pathway in inflammation, and its dysregulation has been implicated in various inflammatory and autoimmune diseases. Moreover, A20 was involved in all five of the top sex-dimorphic hallmark pathways distinguishing male and female control neutrophils, including TNF-α signaling via NF-κB, interferon-gamma response, KRAS signaling up, complement activation, and hypoxia (Fig. 8b). Next, to validate our findings, we assessed the mRNA expression levels of these genes using qRT-PCR. Interestingly, we observed a significant decrease in the expression of *Ly86* and *Ctsg*, alongside a significant increase in *A20* expression, in neutrophils from Zika-exposed male offspring compared to controls (Fig. 8c). In contrast, no significant differences in the expression of these genes were detected in neutrophils from Zika-exposed female offspring (Supplementary Fig. 7). Furthermore, we identified A20 as a sex-dimorphic gene in neutrophils, with significantly higher mRNA expression in female control compared to male control neutrophils (Fig. 8c). Given the significant upregulation of A20 in neutrophils from ZIKV-exposed male offspring, we next investigated whether A20 overexpression could compromise neutrophil activation.

Firstly, we overexpressed A20 in wild-type male neutrophils through transfection and confirmed a significant increase in A20 expression at both mRNA and protein levels (Fig. 8d). Next, we collected conditioned media from LPS-treated neutrophils with or without A20 overexpression after 24 h of stimulation and analyzed cytokine levels using Luminex. Notably, G-CSF, TNF-α and MIP-1α levels were significantly reduced in A20-overexpressing neutrophils following LPS stimulation (Fig. 8e). This reduction was not attributable to the cell death, as 24 h of LPS treatment did not induce substantial neutrophil death (Supplementary Fig. 8). To further assess neutrophil activation, we performed flow cytometry. As shown in Fig. 8f, A20 overexpression alone led to an increased percentage of CD62L$^{+}$ neutrophils and elevated CD62L mean fluorescence intensity (MFI), indicating a less activated neutrophil state without stimulation. Upon LPS stimulation, A20-overexpressing neutrophils exhibited significantly reduced activation, as indicated by decreased CD62L shedding

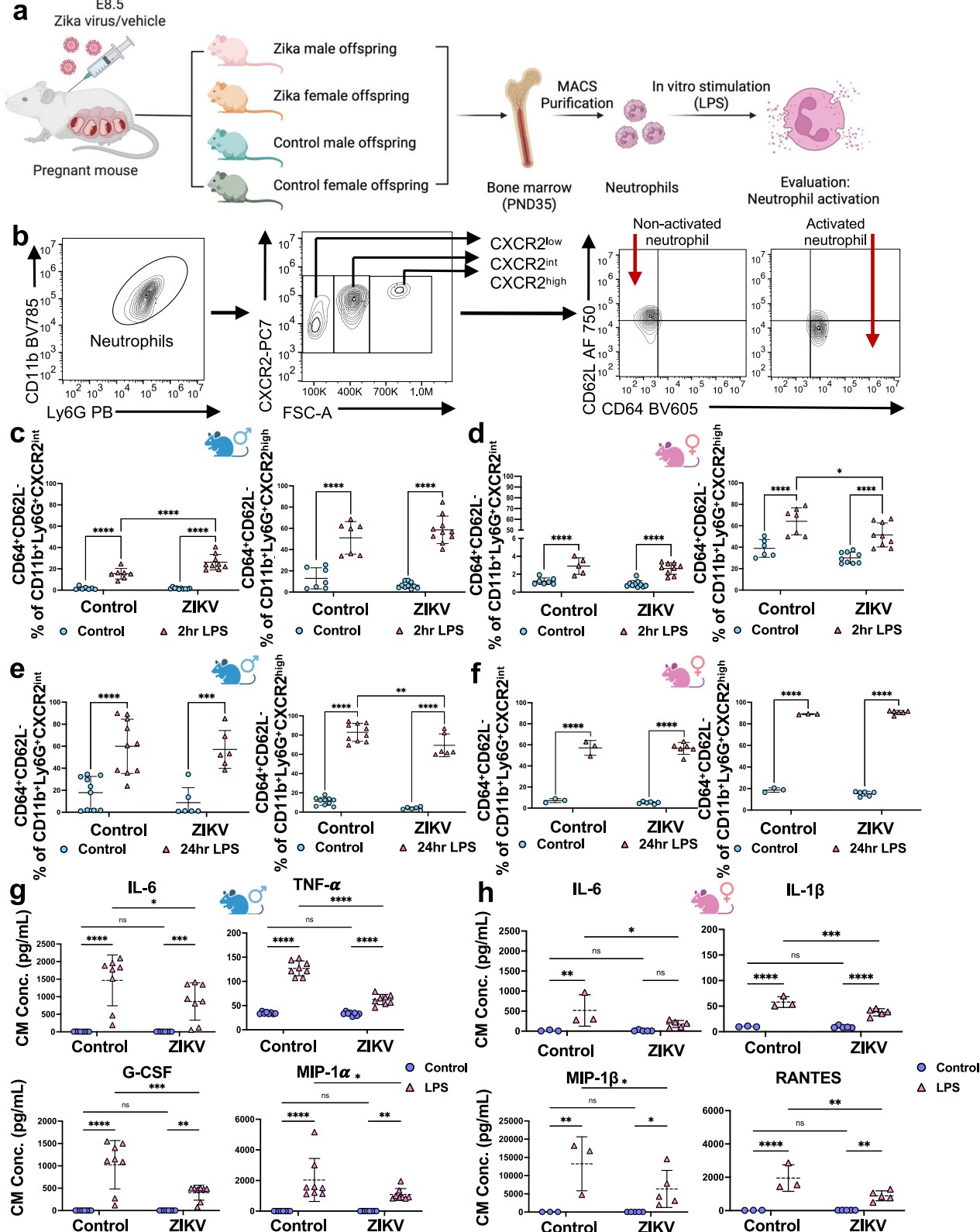

compared to wild-type neutrophils. This suggests that A20 plays an inhibitory role in neutrophil activation during the inflammatory response to LPS.

Next, we investigated the role of A20 in ROS production, processes disrupted by *in utero* ZIKV infection according to our data. First, we measured ROS levels in neutrophils with or without A20 overexpression following PMA stimulation. The results demonstrated that

A20 overexpression did not alter ROS production in response to PMA treatment (Fig. 8g).

To explore the role of A20 in regulating neutrophil death, we assessed the impact of A20 overexpression on neutrophil viability using a 4-h PMA stimulation protocol followed by flow cytometry analysis. MPO, a granule-resident enzyme stored in azurophilic granules, reflects granule integrity and neutrophil identity, while Zombie

**Fig. 7 | Sex-dimorphic impact of Maternal Zika Infection on Offspring Neutrophil Response to LPS. a** Experimental design. Created in BioRender. Alvero, A. (2025) http://BioRender.com/750w48l. Bone marrow neutrophils were isolated at PND35 from male and female offspring of control ($n$ = 3–4 isolations) and ZIKV-exposed groups ($n$ = 3–4 isolations) and treated with 100 ng/mL LPS to assess activation. Each isolation pooled neutrophils from 2–3 mice and was plated in duplicate or triplicate. **b** Representative scatter plots show gating of CD11b$^+$Ly6G$^+$ neutrophil subsets. **c**, **d** After 2 h LPS, the percentage of activated neutrophils (CD64$^+$CD62L$^-$) within CXCR2$^{int}$ and CXCR2$^{high}$ populations was analyzed by sex: control untreated and LPS groups had $n$ = 3 isolations, ZIKV untreated and LPS groups had $n$ = 4 isolations. *$p$ = 0.0142, ****$p$ < 0.0001. **e**, **f** After 24 h LPS, activated neutrophils were analyzed: for males, control untreated and LPS ($n$ = 4 isolations), ZIKV untreated and LPS ($n$ = 3 isolations); for females, control untreated and LPS

($n$ = 1 isolation), ZIKV untreated and LPS ($n$ = 2 isolations). **$p$ = 0.0021, ***$p$ = 0.0001, ****$p$ < 0.0001. **g**, **h** Cytokine levels in conditioned media after 24 h LPS (Luminex) were measured with male groups $n$ = 3 isolations each, female controls $n$ = 1 isolation, and ZIKV $n$ = 2 isolations. For males, IL-6 (*$p$ = 0.0122, ***$p$ = 0.0007), TNF-α (****$p$ < 0.0001), G-CSF (**$p$ = 0.0087, ***$p$ = 0.0001), and MIP-1α (**$p$ = 0.006, *$p$ = 0.0144) showed significant changes. For females, IL-6 (*$p$ = 0.0175, **$p$ = 0.0033), IL-1β (***$p$ = 0.0006, ****$p$ < 0.0001), MIP-1β (**$p$ = 0.0024, *$p$ = 0.036 for ZIKV control vs. LPS, *$p$ = 0.0447 for LPS control vs. ZIKV), and RANTES (**$p$ = 0.0029 for ZIKV control vs. LPS, **$p$ = 0.0019 for LPS control vs. ZIKV, ****$p$ < 0.0001) were significant. Data are mean ± SD. Two-way ANOVA with Šídák's multiple comparisons was used. Female and male mice symbols (panel **c**–**h**) were created in BioRender. Alvero, A. (2025) http://BioRender.com/u0w40ys. Source data are provided as a Source Data file.

dye marks cells with compromised membrane integrity, serving as an indicator of cell death. In the absence of PMA stimulation, A20 over-expression led to a decrease in the MPO$^+$ Zombie$^+$ population and a corresponding increase in the MPO$^+$ Zombie$^-$ population, suggesting that A20 inhibits spontaneous neutrophil death in vitro (Fig. 8h). Consistently, A20-overexpressing neutrophils exhibited higher CD62L expression under basal conditions (Fig. 8f), indicating a less activated phenotype. Upon PMA stimulation, however, no significant differences were observed between wild-type and A20-overexpressing neutrophils in either the MPO$^+$ Zombie$^-$ or MPO$^+$ Zombie$^+$ populations (Supplementary Fig. 9), indicating that A20 did not modulate PMA-induced cell death. Together, these findings suggest that A20 promotes neutrophil survival and maintains a less activated state under basal conditions, but does not modulate cell death in response to PMA stimulation.

## Discussion

In the present study, we demonstrate that an anti-viral immune response during pregnancy can differentially shape the immune system of the offspring in a sex-specific manner. We demonstrate that ZIKV infection during pregnancy impacts the normal development of offspring neutrophils, leading to differential responses of the offspring to postnatal bacterial challenges. Furthermore, we have identified A20, as a sex-specific immune regulatory protein that is highly responsive to maternal inflammatory signals and plays a critical role in modulating neutrophil function in the offspring.

Historically, mild/asymptomatic viral infections during pregnancy have long been considered benign conditions, with the maternal systemic immune response to the virus being transient and the virus unable to cross the placenta barrier reaching the fetus[43]. A recent study demonstrated that mild/asymptomatic COVID-19 infection in pregnant mothers triggers a robust immune response in the placenta and altered the transcriptome and function of fetal immune cells in newborn circulation without direct viral transmission[44]. This finding empathizes that it is the placenta response, rather than viral replication in the fetus, that affects offspring immune development. In our study, we established a chronic placental inflammation mouse model by injecting a low dose of ZIKV into pregnant mice at E8.5, when the placenta begins to form. Although no viral titers were detected in the placenta four days post-infection, the increase of CD11b$^+$CD45$^+$ myeloid cells in the placenta following ZIKV infection indicates that maternal infection triggers local immune activation within the placenta, promoting recruitment or proliferation of myeloid cells as a first line of defense. Notably, at seven days post-infection, CD11b$^+$CD45$^+$ myeloid cells accumulation persisted in female placentas but not in male placentas, highlighting a sex-dimorphic immune response, which confirms that male and female placentas respond differently to infection and inflammation[13,45].

To better understand the differential response to ZIKV infection, we analyzed the transcriptional signature of the placenta exposed to

ZIKV infection and observed that at the transcriptomic level, male placentas were enriched for pathways related to innate immune responses, including IFN-β signaling, regulation of IL-1β production, and neutrophil activity. In contrast, female-specific processes were primarily associated with tissue organization and metabolic pathways. This aligns with previous findings showing a significant increase in the expression of ISGs specifically in human male placentas from women with SARS-CoV-2 infections, a pattern not observed in their female placenta cohorts[46].

Emerging evidence supports sex-dependent placental immune responses, with differential production of cytokines, chemokines, and antibodies between male and female placentas[45,47,48]. These immune factors can cross the placenta and shape fetal immune programming, potentially contributing to sex-dimorphic disease susceptibility. In our study, we discovered that the ZIKV-exposed male offspring showed significantly decreased growth and heightened inflammation to LPS challenge, as evidenced by elevated cytokine levels and enhanced neutrophil migration and activation—an effect not observed in female offspring. These findings align with previous reports of sex-specific outcomes in offspring flowing prenatal ZIKV exposure. For example, in a rat model, adult offspring prenatally infected with ZIKV exhibited motor deficits in a sex-specific manner and failed to mount a normal interferon response to the viral mimic Poly(I:C) in the spleen[49]. In mice, male offspring born to dams infected with low-dose ZIKV show an increased risk of neurocognitive disorders later in life[50]. In humans, Foo et al. recently reported that children exposed to ZIKV *in utero* exhibited a chronic Th1-biased immune profile at two years of age and showed impaired responses to Th2-biased vaccines[9], raising concerns about the long-term consequences of prenatal ZIKV exposure on immune function. However, this study did not stratify analyses by sex, leaving the potential for sex-dimorphic immune outcomes unaddressed. Our investigation into the exaggerated inflammatory response to LPS in ZIKV-exposed male offspring revealed significant alterations in neutrophil dynamics, including migration and activation. RNA sequencing of bone marrow neutrophils demonstrated that *in utero* ZIKV exposure led to sex-dimorphic changes in gene expression in offspring neutrophils. Functional assays further showed that both ROS production and NET formation were significantly reduced in male and female offspring, despite RNA-seq data indicating a more pronounced transcriptional effect in male neutrophils. Specifically, DEGs associated with NETs formation included *H2ac11, H2az2, H2ax, H4c14* and *H2ac13*, and these histones such as H2AX, H2AZ2, and H4 family members play a critical role in chromatin decondensation[51,52], a prerequisite for NETosis[53]. Therefore, it is possible that epigenetic regulation induced by *in utero* ZIKV exposure, may underlie the impaired NETosis observed in offspring. Further investigation is warranted to determine whether specific histone modifications, such as acetylation, methylation, or phosphorylation, contribute to the observed sex-specific differences in offspring neutrophil function.

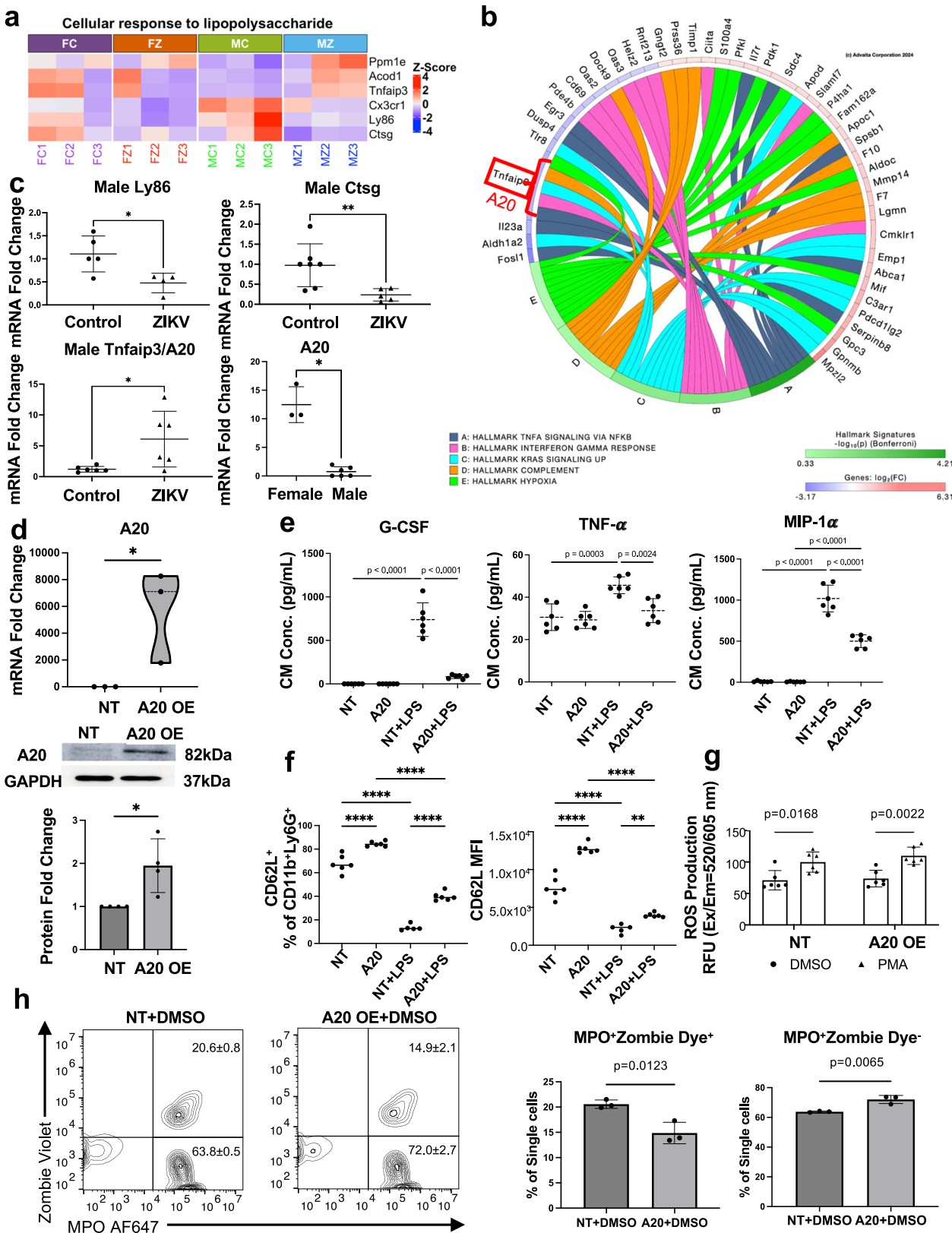

Notably, we observed a significantly reduced cytokine production in isolated neutrophils from ZIKV-exposed offspring following LPS stimulation. Firstly, baseline cytokine profiles differed between male and female neutrophils. In response to LPS challenge, IL-6, TNFα, G-CSF, and MIP-1α were significantly elevated in male neutrophils, whereas female neutrophils exhibited increased IL-6, IL-1β, MIP-1β, and

RANTES. However, following *in utero* ZIKV exposure, this LPS-induced cytokine response was blunted in both sexes, consistent with previous findings in ZIKV-exposed porcine peripheral blood mononuclear cells (PBMCs) showing elevated basal IL-1β but no increase to LPS challenge ex vivo[54]. Transcriptomic analysis revealed downregulation of *Ly86* and *Ctsg*, along with upregulation of *A20* in male neutrophils. *Ly86* encodes

**Fig. 8 | Role of A20 in regulating neutrophil function. a** Heatmap of DEGs related to "Cellular response to lipopolysaccharide." **b** Chord diagram illustrating Tnfaip3/A20 involvement in the top five sex-dimorphic hallmark pathways distinguishing male and female control neutrophils. **c** Validation of Ly86, Ctsg, and Tnfaip3/A20 mRNA expression in male mouse neutrophils by qRT-PCR. $n = 5$ control, 4 ZIKV for Ly86 ($p = 0.02$); $n = 7$ control, 5 ZIKV for Ctsg ($p = 0.0097$); $n = 6$ each for Tnfaip3/A20 ($p = 0.025$); for A20 sex dimorphism, female neutrophils $n = 3$, male neutrophils $n = 6$ ($p = 0.0238$). $n$ denotes independent isolations pooled from 2–3 mice and run in duplicate for qRT-PCR. **d** A20 overexpression in primary male bone marrow neutrophils confirmed by qRT-PCR ($n = 3$ independent experiments, $p = 0.0227$) and Western blot ($n = 4$ independent experiments, $p = 0.0459$). Neutrophils were pooled from 2–3 mice in each independent experiment. **e** Cytokine levels in conditioned media after 24 h of LPS treatment, measured by Luminex. $n = 6$, neutrophils isolated from 3 mice and plated in duplicate. **f** Percentage of

CD62L$^+$ cells within CD11b$^+$Ly6G$^+$ neutrophils and CD62L MFI after 24 h of LPS treatment, with or without A20 overexpression. $n = 6$, neutrophils isolated from 3 mice and plated in duplicate except NT LPS ($n = 5$ due to sample loss). CD62L MFI: NT LPS vs. A20 LPS ($p = 0.0029$), ****$p < 0.0001$. **g** ROS production of isolated neutrophils following PMA treatment (45 min) with or without A20 overexpression. $n = 6$, neutrophils isolated from 2 mice and plated in triplicates. **h** Representative scatter plot and percentage of non-viable neutrophils (MPO$^+$Zombie Dye$^+$) and viable neutrophils (MPO$^+$Zombie Dye$^-$). $n = 3$, one isolation from 2 mice and plated in triplicates. Data are presented as mean ± SD. Two-sided Student's $t$-test (**c**, **d**, **h**), one-way ANOVA with Tukey's test (**e**, **f**), and two-way ANOVA with Šídák's multiple comparisons (**g**) were used. FC female control neutrophils, FZ female ZIKV neutrophils, MC male control neutrophils, MZ male ZIKV neutrophils, NT no treatment, OE overexpression. Source data are provided as a Source Data file.

a protein that may function in concert with TLR4 to mediate innate immune responses and cytokine production[40], while *Ctsg* (cathepsin G) is a serine protease involved in modulating inflammation by cytokines and chemokines[41]. The downregulation of these genes may contribute to the diminished cytokine production observed in male neutrophils following LPS stimulation. Remarkably, these gene expression changes were not observed in female neutrophils, suggesting that the mechanisms underlying the attenuated inflammatory response to LPS are sex-specific. Further investigation is needed to elucidate the distinct regulatory pathways driving this dimorphism.

Interestingly, *Tnfaip3* (A20), a central negative regulator of NF-κB signaling[42] implicated in various inflammatory and autoimmune diseases[55–57] was strongly upregulated in male neutrophil following *in utero* ZIKV exposure. Baseline expression was higher in female neutrophils but increased specifically in males after exposure. Given its role in suppressing NF-κB signaling[42], this upregulation likely contributes to the attenuated inflammatory response to LPS observed in ZIKV-exposed males. Previous studies have demonstrated that A20 plays a critical role in regulating neutrophil recruitment and inflammatory responses[58–60]. In our study, overexpression of A20 suppressed neutrophil activation in response to LPS but did not affect ROS production or NET formation in response to PMA, suggesting that the impaired NET formation in ZIKV-exposed offspring is A20-independent and may involve other regulatory mechanisms that require further investigation. Furthermore, neutrophils are short-lived cells that undergo spontaneous death as a key mechanism to maintain immune homeostasis and prevent excessive inflammation[61]. A20 is a well-known regulator of this process, acting as a ubiquitin-editing enzyme that controls inflammatory signaling and cell survival[62,63]. In our study, A20 overexpression inhibited spontaneous neutrophil death while limiting neutrophil activation under basal conditions, consistent with previous findings that A20 modulates neutrophil death by regulating NF-κB signaling and suppressing pro-apoptotic pathways[63–65]. These findings highlight A20's crucial role in balancing neutrophil survival and activation. Further studies are necessary to better understand what differentially regulates A20 function in male but not in female neutrophils following *in utero* Zika exposure.

In summary, our study characterizes the sex-dimorphic effects of chronic placental inflammation induced by ZIKV infection during pregnancy on offspring neutrophil function and infection susceptibility. Notably, even offspring that appear healthy at birth may exhibit dysregulated neutrophil responses upon secondary immune challenges, such as infections, underscoring the need for long-term monitoring of ZIKV-exposed children. While our findings focus on ZIKV as a model pathogen, they have broader implications for understanding how *in utero* viral exposures shape immune development and offspring health. Future research should aim to longitudinally assess neutrophil function in virus-exposed children to determine whether similar immune dysregulation persists in humans. Ongoing studies in our laboratory aim to identify placental signaling pathways that is

responsible for programming offspring neutrophil function, providing critical insights into the mechanisms underlying infection-driven immune reprogramming during fetal development.

## Methods
### RNA sequencing
Sample RNAs were extracted using Qiagen RNeasy Mini kit following manufacturer's instructions (Qiagen, Hilden, Germany). RNA sequencing of placenta and neutrophil were performed at GENEWIZ LLC (South Plainfield, New Jersey, USA). Library preparation RNA sequencing was performed using the NEBNext Ultra II RNA Library Prep Kit for Illumina according to manufacturer's instructions (New England Biolabs, Ipswich, MA, USA). The prepared sequencing libraries were validated on the Agilent TapeStation (Agilent Technologies, Palo Alto, California, USA). Libraries were quantified using the Qubit 2.0 Fluorometer (ThermoFisher Scientific, Waltham, Massachusetts, USA) and by quantitative PCR (KAPA Biosystems, Wilmington, Massachusetts, USA). Sequencing libraries were multiplexed and clustered on flow cell and then loaded onto the Illumina HiSeq X Ten instrument according to manufacturer's instructions. Sequencing was conducted using a 2x150bp Paired-End configuration for the samples by GENEWIZ. Finally, the HiSeq Control Software or NovaSeq Control Software was used for image analysis and base calling for the sample RNA sequences. The raw sequence data (.bcl files) generated from Illumina instruments were converted into fastq files and de-multiplexed using Illumina bcl2fastq 2.17 conversion software.

### Bioinformatic analysis
**Quality control and pre-processing.** Quality of the raw sequencing data was investigated, and sequence reads were trimmed to remove adaptor sequences and low-quality bases using Trimmomatic v0.36.88. The trimmed reads were mapped onto the mouse reference genome (GRCm39) available on GENCODE (release M34) using Hisat2 v2.2.1, unique gene hit counts were calculated with using the featureCounts function from the Subread package v1.5.2.90. Only unique reads which fell within exon regions were counted. Gene expression data were analyzed as described below.

**Differential gene expression analysis.** Normalization and differential expression analysis were performed using DESeq2 (v1.36.0) in R. Genes with absolute log2 fold-change (abs(Log2FC)) > 0.6 and raw $p$-value < 0.05 were considered differentially expressed.

**Gene ontology (GO) enrichment analysis.** To determine enriched biological processes, GO term analysis was performed using the iPathwayGuide software from AdvaitaBio (v18.1). Significantly enriched biological processes (adjusted p-value < 0.05) were identified for male- and female-specific DEGs. DEGs associated with the biological functions indicated by iPathway were used to create heatmaps, with each function labeled at the top of the respective heatmap. *Data*

*visualization:* Heatmaps of DEGs were plotted using ComplexHeatmap (v2.22.0). PCA plots were visualized using PCAtools (v2.20.0). Venn diagrams comparing biological processes and pathways were generated using the iPathwayGuide software from AdvaitaBio. Dot plots for pathway enrichment were created using ggplot2 (v3.4.0). Unique and shared pathways between male and female placental responses were visualized using Venn diagrams and dot plots. Dot plots of enrichment factors and false discovery rate (-log10(FDR)) were created using ggplot2 (v3.4.0).

## Mouse model

Female and male C57BL/6 mice were obtained from the Jackson Laboratory (Bar Harbor, ME) and bred in a specific-pathogen-free facility at Wayne State University. Animals were housed under standard laboratory conditions with a 12-h light/12-h dark cycle, controlled temperature ($22 \pm 2\,°C$) with 40–60% relative humidity, and ad libitum access to food and water. All experiments involving ZIKV were conducted in a Biosafety Level 2+ (BSL-2+) facility following institutional and federal guidelines to ensure safety and containment. Adult (8–10 weeks old) C57BL/6 mice were set up for timed mating, with the presence of a vaginal plug designated as embryonic day (E) 0.5. On E8.5, plug-positive mice were randomly assigned to receive an intraperitoneal (i.p.) injection of either 50 PFU of Zika virus (in $100\,\mu L$) or vehicle control (1% FBS DMEM/F12). This low-dose, systemic exposure model was designed to mimic the mild/asymptomatic maternal viral infections commonly observed during pregnancy, which often go undetected but can have significant consequences for the fetus. Intraperitoneal (i.p.) injection was chosen to ensure consistent systemic exposure to viral antigens across animals and to initiate a uniform immune stimulus, which can contribute to downstream placental immune activation even in the absence of detectable maternal viremia. Pregnant mice were euthanized on E12.5 or E17.5 for assessment of pregnancy outcomes and tissue collection. Placenta sexes were determined by qRT-PCR for specific Y chromosome-dependent gene expression (*Ddx3y*). For offspring studies, pregnant females were allowed to deliver naturally, and offspring were randomly assigned for experiments at postnatal day 35–40 (PND35-40). Investigators were blinded to group allocation during data collection and analysis by coding samples. All procedures were performed in accordance with National Institutes of Health guidelines (NIH Publication No. 85–23, revised 1996) and approved by the Institutional Animal Care and Use Committee (IACUC) at Wayne State University under protocol number 25-02-7578. Sex was included as a biological variable. Both female and male mouse offspring were studied and analyzed separately to assess sex-specific effects.

## Virus

ZIKV strain FSS13025, originally isolated in Cambodia in 2010, was obtained from the World Reference Center for Emerging Viruses and Arboviruses at University of Texas Medical Branch, Galveston[66]. ZIKA virus was propagated in African green monkey kidney (Vero) cells by infecting monolayers with viral stock. Once a complete cytopathic effect was observed, the infected supernatant was collected, centrifuged, aliquoted, and stored at $-80\,°C$. The infectious viral titers of the viral stocks were determined by plaque assay.

## Virus titer quantification by qRT-PCR

ZIKV titer was quantified using one-step quantitative reverse transcriptase PCR (qRT-PCR). Viral RNA was extracted from ZIKV stock using the QIAamp MinElute Virus Spin Kit (Qiagen, catalog no. 57704) and used to generate a standard curve through serial tenfold dilutions. One microgram of RNA from each sample was analyzed on a CFX96 C1000 qPCR system (Bio-Rad) using a one-step PCR mix (Promega, catalog no. A6120). Zika virus was detected using the following primer pair: forward 5′-CCGCTGCCCAACACAAG-3′ and reverse 5′-AGCCT ACCTTGACAAGCAGTCAGACACTCAA-3′. The probe sequence is 5′-AGCCTACCTTGACAAGCAGTCAGACACTCAA-3′ 6- FAM/ZEN/IBFQ[67,68]. The cycling conditions included reverse transcription at $45\,°C$ for 15 min and initial denaturation at $95\,°C$ for 2 min, followed by 40 cycles of $95\,°C$ for 15 s and $60\,°C$ for 1 min. Viral RNA levels were quantified by comparing sample threshold cycle (Ct) values to a ZIKV RNA standard curve.

## Protein extraction and Western blotting

For protein extraction, neutrophils were lysed on ice in cell lysis buffer (1% Triton X-100, 0.05% SDS, 100 mM Na2PO4, and 150 mM NaCl) supplemented with protease inhibitor mixture (Roche) and PMSF for 15 min followed by centrifugation at $16,000 \times g$ for 15 min at $4\,°C$ to remove cell debris. The protein concentration was determined by bicinchoninic acid (BCA) assay (Pierce, catalog no. 23223, Rockford, IL). $10\,\mu g$ of each protein lysate was electrophoresed on a 12% SDS–polyacrylamide gel and transferred onto polyvinylidene difluoride (PVDF) membranes (EMD Millipore) at 32 V overnight. The membranes were blocked with PBS-0.05% Tween 20 (PBS-Tween) containing 5% nonfat milk (Fisher Scientific, Pittsburgh, PA), washed three times, and incubated with primary antibody in 1% milk PBS-Tween at $4\,°C$ overnight. After three additional washes in PBS-Tween, membranes were incubated with the appropriate secondary antibody in 1% milk PBS-Tween for 2 h at room temperature. Immunoreactivity was detected using enhanced chemiluminescence (NEN Life Sciences, Waltham, MA) and imaged by ImageQuant LAS 500 (GE Healthcare). Antibodies were diluted as follows: 1:1000 anti-A20 (Cell Signaling Technology, catalog no. 5630); 1:10,000 anti-GAPDH (Sigma, catalog no. G8795); 1:10,000 anti-β-Actin (Cell Signaling Technology, catalog no. 4967); and 1:10,000 peroxidase-conjugated anti-rabbit IgG (Cell Signaling Technology, catalog no. 7074); and 1:10,000 peroxidase-conjugated anti-mouse IgG (Cell Signaling Technology, catalog no. 7076). Western blot band intensities were quantified using ImageJ software 1.49v (NIH, USA). Uncropped and unprocessed scans of Western blots are provided in the Source Data file.

## RNA extraction and quantitative PCR

Total RNA was extracted from neutrophils using the RNeasy Mini Kit (Qiagen, catalog no. 74106) following the manufacturer's instructions. For tissue RNA extraction, samples were homogenized in TRIzol (Ambion, Waltham, MA, USA) using 1.0-mm zirconium beads (Benchmark, Sayreville, NJ, USA) for two cycles at $400 \times g$ for 2 min. The homogenate was incubated at room temperature (RT) for 5 min, followed by the addition of $200\,\mu L$ chloroform per mL of TRIzol. Samples were vortexed for 15 s, incubated for 3 min at RT, and centrifuged at $12,000 \times g$ for 15 min at $4\,°C$. The aqueous phase was transferred to a new tube, mixed with $500\,\mu L$ of 100% isopropanol, incubated at RT for 10 min, and centrifuged at $12,000 \times g$ for 10 min at $4\,°C$. The supernatant was discarded, and the pellet was washed twice with $500\,\mu L$ of 100% ethanol, followed by two washes with 1 mL of 75% ethanol. After vertexing and centrifugation at $7500 \times g$ for 5 min at $4\,°C$, the pellet was air-dried for 10 min and resuspended in $50\,\mu L$ of RNase-free water. RNA concentration and purity were assessed using spectrophotometric analysis (A260/A280), with only samples displaying a ratio of ≥1.8 used for PCR. One microgram of RNA was reverse-transcribed using the iScript cDNA Synthesis Kit (Bio-Rad, Hercules, CA, USA). Quantitative PCR was performed using iTaq Universal SYBR Green Supermix (Bio-Rad) and gene-specific primers (Supplementary Table 3) on a CFX96, C1000 system qPCR machine (Bio-Rad). Gene expression was normalized to GAPDH and analyzed using the $2^{-\Delta\Delta Ct}$ method[69].

## Cytokine analysis

Cytokine concentrations were measured using the Bio-Plex Pro Mouse Cytokine 23-plex Assay (Bio-Rad, #60009RDPD), which included IL-1α,

IL-1β, IL-2, IL-3, IL-4, IL-5, IL-6, IL-9, IL-10, IL-12 (p40), IL-12 (p70), IL-13, IL-17A, Eotaxin, G-CSF, GM-CSF, IFN-γ, KC, MCP-1, MIP-1α, MIP-1β, RANTES, and TNF-α. Assays were performed according to the manufacturer's protocol. Briefly, fluorescently labeled capture beads were loaded into a 96-well magnetic plate and incubated with 50 μL of either standards, cell-free supernatants, or fourfold diluted serum samples on an orbital shaker (500 rpm) for 30 min at room temperature in the dark. Wells were washed three times by adding 100 μL wash buffer, mixing briefly, and removing the buffer by inverting the plate. Beads were then incubated with 25 μL biotinylated detection antibodies under the same conditions. After three additional washes, 50 μL streptavidin-phycoerythrin (PE) was added and incubated for 10 min at 500 rpm in the dark. Following a final wash, beads were resuspended in 125 μL assay buffer and analyzed using a LUMINEX 200 system (LUMINEX, Austin, TX). Data acquisition and analysis were performed using LUMINEX xPONENT software v4.3.

### Neutrophil ROS production
ROS production by mouse neutrophils was quantified using the Cellular ROS Assay Kit (Red) (Abcam, Cat. No. ab186027) according to the manufacturer's protocol. Freshly isolated mouse neutrophils ($5 \times 10^4$ cells in 100 μL of neutrophil culture medium: RPMI 1640 without phenol red, 1% penicillin/streptomycin, 10% BSA) were seeded in a 96-well plate and incubated overnight at 37 °C with 5% $CO_2$. On the following day, the red ROS detection dye was added to the wells and incubated for 1 h at 37 °C with 5% $CO_2$. Neutrophils were then stimulated by adding 20 μL of 11× PMA (550 nM) in PBS or vehicle control (DMSO) to each well, followed by a 45-min incubation at 37 °C with 5% $CO_2$. Fluorescence intensity was measured using a fluorescence microplate reader at Ex/Em = 520/605 nm (cut-off 590 nm) in bottom-read mode. Background fluorescence was subtracted from all measurements, and relative fluorescence units (RFU) were compared across groups. Each condition was assessed in triplicate.

### Neutrophil isolation and NETs staining
Mouse bone marrow neutrophils from control and Zika-exposed offspring were freshly isolated from the tibias and femurs using the Neutrophil Isolation Kit (Miltenyi Biotec, Cat#130-097-658) according to the published protocol[70]. The purity of the isolated neutrophils was assessed by flow cytometry using CD11b and Ly6G staining, with purity exceeding 95%. $1.5 \times 10^5$ neutrophils were seeded in 150 μL of neutrophil culture medium (RPMI 1640 without phenol red, supplemented with 1% penicillin/streptomycin and 10% BSA) in a 48-well plate. SYTOX Green (1:100 dilution) was added to the culture medium to label extracellular DNA. Neutrophils were stimulated with phorbol 12-myristate 13-acetate (PMA, 40 nM) in PBS or vehicle control (DMSO), and NET formation was monitored over time. At 48 h post PMA treatment, when the maximal amount of NETs was observed, cells and NETs were fixed with 4% paraformaldehyde for 15 min at room temperature, followed by three washes with cold PBS. Cells were then permeabilized with 0.4% Triton X-100 for 15 min, washed three times with cold PBS, and blocked with 1% BSA in PBST for 1 h at room temperature. After three PBS washes, cells were incubated overnight at 4 °C with a primary anti-myeloperoxidase (MPO) antibody (R&D, Cat# AF3667, 6.5 μg/mL). The following day, cells were washed three times in cold PBS and incubated with a secondary antibody (Alexa Fluor 546 rabbit anti-goat, 1:500 dilution, Cat# A-21085, ThermoFisher Scientific) for 1 h at room temperature, followed by three PBS washes. Images were acquired using an Echo Revolve Microscope (Echo Laboratories, San Diego, CA, USA). Co-localization of SYTOX Green and MPO confirmed the presence of NETs.

### A20 overexpression
Neutrophils ($2–3 \times 10^6$) were seeded in a 12-well plate in 500 μL of neutrophil culture medium (RPMI 1640 without phenol red,

supplemented with 1% penicillin/streptomycin and 10% BSA). For transfection, a mixture was prepared containing 6 μL of Escort III transfection reagent (Sigma, #L3037), 2 μg of mouse A20 overexpression plasmid (Origene, #MR210582), and 91.5 μL of Opti-MEM. The mixture was incubated at room temperature for 20 min in a biosafety hood before being added dropwise to the wells. Plates were gently swirled to ensure even distribution. After 5 h, cells were centrifuged at $1500 \times g$ for 5 min, the medium was removed, and fresh neutrophil culture medium (1 mL) was added. Cells were then incubated at 37 °C with 5% $CO_2$ for an additional 19 h before collection or further treatment.

### Flow cytometry
Cell surface staining was performed in FACS Buffer (1X PBS + 1% bovine serum albumin + 0.05% sodium azide) and intracellular staining was performed by fixing and permeabilizing using BD Cytofix/Cytoperm (#554714) kits following the manufacturer's protocol. Data were acquired using CytoFLEX analyzer (RRID: SCR_019627) and CytExpert v2.6 (RRID: SCR_017217) acquisition software (Beckman Coulter, Brea, CA). Data were analyzed using FlowJo v10.10.0 (RRID: SCR_008520; Becton, Dickinson and Company, Ashland, OR). Gating strategy for the analysis of samples is shown in Supplementary Fig. 10. Cell surface and intracellular staining was performed using the following fluorophore-conjugated antibodies (BioLegend, unless indicated otherwise): CD11b (M1/70, #101243), Ly6G (1A8, #127612), CD45 (30-F11, #103114), CXCR2 (SA045E1, #149618), CD101 (307707, #564473, BD Biosciences), CD62L (MEL-14, #104450), CD64 (X54-5/7.1, #139323), MPO (8F4/MPO, #570233, BD Biosciences), Zombie Violet Fixable Viability Kit (#423113).

### Statistical analysis
Statistical analyses were conducted using GraphPad Prism software, version 10.4.0 (San Diego, CA, USA). Data are presented as mean ± standard deviation (SD). Outliers (data points >2 standard deviations from the mean) were excluded. This exclusion criterion was pre-established to reduce the influence of extreme values on statistical outcomes. Normality was assessed using the Shapiro–Wilk test. If data did not follow a normal distribution, they were either transformed (log or square-root) or analyzed using non-parametric tests. Homogeneity of variance was evaluated using Levene's test. For comparisons between two groups, two-sided Student's $t$-test was performed. One-way ANOVA was used when analyzing differences among multiple groups with a single independent variable, such as different treatment conditions. Two-way ANOVA was applied to assess the interaction between two independent factors, such as sex and treatment. When significant main effects or interactions were detected, post-hoc comparisons were conducted using Tukey's test or Šídák's multiple comparison test to determine specific group differences. A $p$-value < 0.05 was considered statistically significant.

### Reporting summary
Further information on research design is available in the Nature Portfolio Reporting Summary linked to this article.

## Data availability
The raw and processed RNA sequence data form our mouse placentas and neutrophil experiments are available in NCBI's Gene Expression Omnibus under the accession number GSE292966. Source Data are provided with this paper and deposited in Figshare at https://doi.org/10.6084/m9.figshare.29538431.

## Code availability
Gene Ontology and pathway analyses were performed using the licensed iPathwayGuide software (AdvaitaBio). The R scripts include steps reliant on this licensed software and contain file paths specific to

our local computing environment. Therefore, the scripts are available upon reasonable request from the corresponding author (email: candy.ding@wayne.edu) for non-commercial academic research purposes. Requests will be processed within two weeks, and scripts will be available for at least five years after publication.

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

## Acknowledgements

We gratefully acknowledge the contribution of the animals used in this study. We thank Alex Fox for expert assistance with flow cytometry, Bin Wu for support with animal dissection and neutrophil isolation, and Susan Wissman for dedicated animal care. This work was supported in part by NIH grant NIAID 5R01AI145829-05 (GM), 5P42ES030991 (GM Project 4), R01HD111146 (GM) and R00ES028734-01A1 (MP).

## Author contributions

Conceptualization, J.D.; Methodology, J.D., A.H., A.N., D.M., E.F., and A.M.; RNA sequencing data analysis, G.S., A.S., and N.A.; Writing- Original Draft, J.D.; Writing-Review & Editing, J.D., Y.H., M.G., and G.M.; Supervision, G.M. and J.D.; Funding Acquisition, G.M.

## Competing interests

The authors declare no competing interests.
