## [Transparent Peer Review file · Nature Communications]

Prenatal exposure to Zika virus shapes offspring neutrophil function in a sex-specific manner

Corresponding Author: Dr Jiahui Ding

Version 0:

Reviewer comments:

Reviewer #1

(Remarks to the Author)

This manuscript describes how prenatal exposure to zika virus infection shapes offspring development and neutrophil function in a sex specific manner. Of high interest is the finding that female offspring with prenatal ZIKV infection has fewer developmental or immunological abnormalities compared to male offspring even within the same litters. I think the topic is of high interest and timely. Several revisions of the manuscript that solidify the male vs female differences should be made to improve the manuscript.

Results:

Figure 1. The use of pink for both females (1B, E, F) and ZIKV infected mice (A, C, D) is confusing. It would be better to visualize the ZIKV infected data in a distinct color from the males and females.

Figure 1E, F: besides placental weight and fetal weight please also depict the placental/fetal weight ratio as it has value for the level of placental function/dysfunction.

Figure 1J: the weight difference of control and ZIKV infected males seems very small with a wide range of weights. Can the authors comment on the biological relevance of this weight range as well as the weight difference?

Figure 2C, D and lines 188-190: The figure depict the comparison of control vs ZIKV infected mice separated by fetal sex, but does not show a direct comparison of male vs female offspring. Please add and discuss this.

Figure 3A: it is unclear why the authors choose to only depict the data separated by male and female offspring. Please include a combined PCA of all male and female controls and ZIKV infected animals to show how similar or different the male and female placenta's are in controls and ZIKV infection.

Figure 3B: similarly please depict the sex specific DEGs in female vs male controls and female vs male ZIKV placentas.

Figure 4A and lines 241-243: this figure doesn't seem to separate male and female offspring and not support the sentence "female control mice mounted a significantly stronger inflammatory response, characterized by elevated production of cytokines such as CXCL1, G-CSF, MCP-1, CCL4, and RANTES". Similarly for Figure 4B and the remaining part of the paragraph specifies sex specific differences that don't seem to be depicted in the figure?

Figure 5: most of the changes seem to be related to ZIKV infection but not due to male or female offspring differences? To solidify the sex based differences please depict the sex comparisons more directly. If there are few, the ZIKV differences should be emphasized as a strong clinically significant finding.

Figure 6: the figure can be improved by more directly depicting male vs female differences. Now the control vs ZIKV is emphasized.

Reviewer #2

(Remarks to the Author)

See attachment.

Version 1:

Reviewer comments:

Reviewer #1

(Remarks to the Author)

The authors have responded to all comments raised by the reviewers and significantly improved the manuscript. I have no further comments.

Reviewer #2

(Remarks to the Author)

The authors have addressed each of my comments and revised the manuscript to improve clarity and ensure to be coherent the presented data. I have a few minor comments:

Line 260: The sentence beginning with “we used the same model to assess...” is inaccurate and potentially misleading. A more accurate phrasing would be, “we followed the previous protocol (21) with modifications,” since the dose, route, and timing of virus administration during pregnancy in this study differ from the conditions described in citation 21. In the study by Szaba et al., the lowest dose examined was 3.4×10^4 PFU, at which fetal growth was comparable to that of mock-infected dams. However, that study did not assess the cellular or molecular impacts of ZIKV exposure during pregnancy. In contrast, the current study intriguingly shows that a much lower dose (50 PFU) can elicit a placental immune response, leading to sex-specific neutrophil response—an effect that could easily be overlooked.

Line 342: Use “unique” instead of “uniquely,” and add the word “processes” after “4.”

Line 343: Use the full term “interleukin-1 beta (IL-1 β)” on first mention. Then, use the abbreviation “IL-1 β ” on line 354 and later.

The same is applicable to interferon-1 β in a few places.

Comments from Reviewer #1

We thank the reviewer for the positive comments about our study. We have incorporated the suggestions, which has improved the quality of the manuscript.

The following are the responses to the specific comments:

1. Reviewer: Figure 1. The use of pink for both females (1B, E, F) and ZIKV infected mice (A, C, D) is confusing. It would be better to visualize the ZIKV infected data in a distinct color from the males and females.

Response: Thank you for the helpful suggestion. We have changed the color representing the ZIKV-infected groups to green to clearly distinguish them from the female and male groups, improving the figure's clarity.

2. Reviewer: Figure 1E, F: besides placental weight and fetal weight please also depict the placental/fetal weight ratio as it has value for the level of placental function/dysfunction.

Response: Thank you for the suggestion. Although no significant differences were observed between groups, we have included the fetal-to-placental weight ratio data in the supplemental material (Supplementary Figure 1) and described these results in the Results section (page 6, line 230-233).

3. Reviewer: Figure 1J: the weight difference of control and ZIKV infected males seems very small with a wide range of weights. Can the authors comment on the biological relevance of this weight range as well as the weight difference?

Response: Thank you for the insightful question. The observed body weight variability among control offspring at postnatal day 35 (PND35), likely reflects normal biological variation influenced by factors such as litter size, and early-life environmental conditions. Variations in litter size can affect nutrient availability and maternal care, which in turn impact individual growth rates. Additionally, PND35 corresponds to the periadolescent

growth phase in mice, during which differences in growth trajectories among individuals are common, even within the same litter. Although the weight difference between control and ZIKV-infected males appears small and accompanied by variability, the statistically significant reduction in weight in the ZIKV-exposed group suggests a biologically relevant effect of in utero ZIKV exposure on postnatal growth.

4. Reviewer: Figure 2C, D and lines 188-190: The figure depict the comparison of control vs ZIKV infected mice separated by fetal sex, but does not show a direct comparison of male vs female offspring. Please add and discuss this.

Response: We apologize for the confusion. Figure 2C illustrates changes in myeloid cell populations in the placenta at E12.5 and E17.5, comparing control and ZIKV-infected groups separated by fetal sex. Figure 2D provides a direct comparison between female and male placentas at each time point, showing a sex- and time-dependent effect: at E12.5, male placentas exhibit a more pronounced increase in this myeloid population compared to females, whereas at E17.5, female placentas demonstrate a more sustained significant increase relative to males. We have clarified the response in the revised version of the manuscript.

5. Reviewer: Figure 3A: it is unclear why the authors choose to only depict the data separated by male and female offspring. Please include a combined PCA of all male and female controls and ZIKV infected animals to show how similar or different the male and female placenta's are in controls and ZIKV infection.

Response: We appreciate the reviewer's concern regarding the presentation of the data separated by male and female placentas. To address this, we have now included a combined principal component analysis (PCA) in Figure 3A that shows all male and female control and ZIKV-infected placentas together. This allows for direct comparison of the similarities and differences between male and female placentas under both control and infection conditions.

6. Reviewer: Figure 3B: similarly please depict the sex specific DEGs in female vs male controls and female vs male ZIKV placentas.

Response: We agree with the reviewer that comparisons of sex-specific differentially expressed genes (DEGs) in female versus male controls and female versus male ZIKV placentas are important for understanding baseline sex dimorphism and responses to viral exposure. To address this, we have included heatmaps for both comparisons in Supplementary Figure 2 and added corresponding descriptions in the manuscript (page 9, lines 353-357).

7. Reviewer: Figure 4A and lines 241-243: this figure doesn't seem to separate male and female offspring and not support the sentence "female control mice mounted a significantly stronger inflammatory response, characterized by elevated production of cytokines such as CXCL1, G-CSF, MCP-1, CCL4, and RANTES". Similarly for Figure 4B and the remaining part of the paragraph specifies 5 of 6 sex specific differences that don't seem to be depicted in the figure?

Response: We apologize for the confusion caused by the presentation of Figure 4. Figure 4A displays the cytokine responses to LPS challenge in male offspring only, at 4 and 24 hours. The corresponding data for female offspring were included in the original supplemental figure, but we recognize this was not clearly indicated in the main text or figure legend.

In the revised version of the manuscript, we have now added an additional supplemental graph (Supplementary Figure 3a) showing the basal cytokine responses to LPS at 4 hours in both male and female offspring, which captures the acute phase of the inflammatory response. This provides clearer support for the sex-specific observations discussed in the text.

Additionally, we have revised the cytokine nomenclature to ensure consistency throughout the paragraph and figure legends. Specifically, we clarified that RANTES is CCL5, MCP-1 is CCL2, and MIP-1 β is CCL4, to prevent confusion due to different alias names.

8. Reviewer: Figure 5: most of the changes seem to be related to ZIKV infection but not due to male or female offspring differences? To solidify the sex based differences please depict the sex comparisons more directly. If there are few, the ZIKV differences should be emphasized as a strong clinically significant finding.

Response: Thank you for the insightful comment. We agree that maternal ZIKV infection during pregnancy has a significant impact on the offspring neutrophil transcriptome. While our data reveal some differences between male and female offspring, the dominant transcriptional changes are driven by ZIKV exposure itself. Therefore, we emphasized the ZIKV-induced alterations as the primary and clinically relevant finding by focusing our analysis and figure presentation on the comparisons between control and ZIKV-exposed groups within each sex, rather than on direct male–female comparisons.

9. Reviewer: Figure 6: the figure can be improved by more directly depicting male vs female differences. Now the control vs ZIKV is emphasized.

Response: Thank you for the helpful comment. Our primary goal in Figure 6 is to emphasize the impact of maternal ZIKV infection on offspring neutrophil function. While baseline sex differences in neutrophils exist, our data show that maternal ZIKV exposure leads to significant immune alterations that are both shared and sex-specific. We intentionally presented the data to highlight how ZIKV exposure, rather than baseline sex differences, drives these changes.

We aimed to show that maternal ZIKV infection does not result in uniform immune outcomes in male and female offspring. Instead, ZIKV induces distinct effects depending on fetal sex. Therefore, we emphasized the comparison between control and ZIKV within each sex group, rather than a direct male versus female comparison, to better reflect the biological impact of maternal infection. We believe this approach more accurately conveys the key message that maternal infection during pregnancy has sex-dimorphic consequences on fetal immune programming.

Comments from Reviewer #2

We thank the reviewer for the positive comments about our study. We have incorporated the suggestions, which has improved the quality of the manuscript.

1. Reviewer: Zika viruses do not establish productive infection in B6 mice. Description of the rationale for the use of B6 mice and the dose of ZIKV (50 PFU) is not clearly stated. Justification of intraperitoneal delivery of virus is not described. Authors provide the information in the study design. Has the impact of ZIKV infection with 50 PFU on maternal immunity in pregnant dams been characterized previously? If so, provide reference for the previous publication or provide data as additional information.

Response: Thank you for this valuable comment. Although ZIKV does not establish productive systemic infection in immunocompetent C57BL/6 mice, we used this low-dose (50 PFU) model to mimic mild or asymptomatic maternal infections during pregnancy, which are commonly observed in clinical settings yet still associated with adverse fetal outcomes. Intraperitoneal (i.p.) injection was chosen to ensure consistent systemic exposure to viral antigens across animals and to initiate a uniform immune stimulus, which can contribute to downstream placental immune activation even in the absence of detectable maternal viremia. While maternal cytokine profiles measured at 4 days post-infection showed no significant systemic inflammation (Supplementary Table 2), both flow cytometry and RNA-seq analyses revealed robust and sustained placental immune responses. These findings support the utility of this model for studying ZIKV-induced placental inflammation and its potential role in shaping fetal immune development. We now include clarification regarding the rationale for this model in the Method Section (Page 29, lines 1372-1378).

2. Reviewer: Some results in Fig2C & D and Fig 3A &B, considering individual variations and heterogeneity among littermate, the sample size, N of 3-5, for transcriptomics is small that may lead to a potentially biased interpretation. Justify the sample size. Are the fetuses from the same single dams or multiple dams? Describe the rationale or limitation of the experimental condition and indicate the number of dams per group in the figure legend

Response: Thank you for this important comment. The placentas used for RNA-seq in Fig. 2c–d and Fig. 3a–b were collected from **2 to 3 dams per group**. To address **intra-litter variation**, we carefully selected one to two male and female placentas from each uterine horn (left and right) per dam. This approach was designed to account for known biological heterogeneity within a litter, including effects of fetal sex, uterine horn side, and intrauterine position, which are known to influence placental development and function (PMID: 40578493, 7799320).

To address **inter-litter variation**, we included placentas from multiple dams per group (2–3), thus capturing variability arising from dam-specific factors such as **maternal physiology, litter size, and immune response to infection**. However, due to limitations in fetal sex distribution, we were unable to perfectly balance sex and horn-side for all samples. As a result, **3–5 biological replicates per group** were selected based on optimal representation and tissue quality.

We acknowledge that the sample size is modest, and increasing it would reduce both intra- and inter-litter variability and improve statistical power. However, funding resource constraints limited our ability to scale the RNA-seq further. Importantly, we have acknowledged this limitation in the Result section (Page 8, line 326-332). Additionally, our validation of key transcriptomic findings using qRT-PCR (**Fig. 3e**), providing additional support for sex-dimorphic responses observed in male and female placentas.

For flow cytometry, placental sex was determined retrospectively, as fresh tissue was required for analysis. We used 2 dams per group at E12.5 and 4 dams per group at E17.5. Due to natural sex ratio differences across litters, the number of male and female samples varies slightly (e.g., fewer female placentas in the E12.5 control group).

These methodological details, including sample selection criteria and the number of dams per group, have been added to the Results section and clarified in the relevant figure legends in the revised version of the manuscript.

3. Reviewer: Line 142-144 (Figure 1), it may be worthwhile to elaborate the impact of ZIKV on a sex -dependent fetal growth in mice and consequences of ZIKV infection in the development of children born from ZIKV exposed pregnant women and discuss. Do boys develop more severe developmental defects than girls in humans? Any clinical reports

regarding the vulnerability of infants/kids born from ZIKV exposed mothers to COVID-19 or other infections?

Response: Thank you for this insightful comment. One human study has reported that male sex is strongly associated with abnormal neurodevelopment in humans following prenatal ZIKV exposure (now cited on page 6, lines 228–230). However, due to limitations in long-term clinical follow-up, there are currently no published studies assessing the susceptibility of infants or children born to ZIKV-exposed mothers to COVID-19 or other infections. We agree this is an important area that warrants future clinical investigation.

Minor Comments:

- **Reviewer:** Line 68-71, the current statement “ZIKV exposure in utero also have long-term consequences...” to maintain subject-verb agreement, change “exposure” to ‘exposures’.

Response: Thanks for pointing out, we have corrected it in the revised version of the manuscript.

- **Reviewer:** Line 268, the statement needs a reference

Response: Thank you. We have added the appropriate reference to support this statement.

- **Reviewer:** Line 280, I interpret the results presented in Fig 4 C and D as showing that LPS challenge stimulates neutrophil activation (an increase of CD62Llo cells), which facilitates the egression of neutrophils from bone marrow into bloodstream. Therefore, the levels of mature, activated neutrophils are transiently reduced in the bone marrow and elevated in the blood circulation. If the authors agree, the current description, “...(Fig. 4D), which was consistent with the significant increased neutrophils in the blood stream...” may be improved as “... coincided with the significantly increase neutrophils....”

Response: Thank you for the helpful suggestion. We have changed the wording to “coincided” for greater accuracy.

- **Reviewer:** Line 410, a typo in ‘time-dependent’

Response: We have edited in the revised version of the manuscript.

- **Reviewer:** Line 417, a typo ‘revealed’ instead of “reveled”

Response: We have edited in the revised version of the manuscript.

- **Reviewer:** Symbols in Fig 6C, Fig 8F and 8G are too small to identify in a print form either use bigger symbols or color coded

Response: We appreciate the reviewer's feedback. We have increased the symbol size to improve readability in print.

- **Reviewer:** Fig8F the y-axis label of CD62L MFI is not legible, log transform the scale would improve readability.

Response: We have enlarged the y-axis label font to enhance readability.

Comments from Reviewer #1

1. Reviewer: The authors have responded to all comments raised by the reviewers and significantly improved the manuscript. I have no further comments.

Response: We sincerely thank the reviewer for their positive feedback and greatly appreciate their time and effort in reviewing our paper.

Comments from Reviewer #2

We thank the reviewer for the positive comments about our study. We have carefully incorporated the suggestions, which have improved the quality of the manuscript. Below are our detailed responses to the specific comments:

1. Reviewer: Line 260: The sentence beginning with “we used the same model to assess...” is inaccurate and potentially misleading. A more accurate phrasing would be, “we followed the previous protocol (21) with modifications,” since the dose, route, and timing of virus administration during pregnancy in this study differ from the conditions described in citation 21. In the study by Szaba et al., the lowest dose examined was 3.4×10^4 PFU, at which fetal growth was comparable to that of mock-infected dams. However, that study did not assess the cellular or molecular impacts of ZIKV exposure during pregnancy. In contrast, the current study intriguingly shows that a much lower dose (50 PFU) can elicit a placental immune response, leading to sex-specific neutrophil response—an effect that could easily be overlooked.

Response: Thank you for the insightful suggestion. We have revised the sentence to clarify the dose, route, and timing of ZIKV infection in our study (Line 156-157). We also added a comparison with previous studies using much higher doses, which similarly demonstrated that the placenta can control infection and prevent vertical transmission in mouse models (Line 164).

2. Reviewer: Line 342: Use “unique” instead of “uniquely,” and add the word “processes” after “4.”

Response: Thank you for the suggestion. We have edited the sentence as suggested (Line 200).

3. Reviewer: Line 343: Use the full term “interleukin-1 beta (IL-1 β)” on first mention. Then, use the abbreviation “IL-1 β ” on line 354 and later. The same is applicable to interferon-1 β in a few places.

Response: Thank you for the helpful comment. We have revised the manuscript to ensure that full terms are used at first mention (e.g., “interleukin-1 beta (IL-1 β),” “interferon-1 beta (IFN-1 β),” and “interferon-stimulated genes (ISGs)”) (Line 201-202), with abbreviations used consistently thereafter (Line 213-214, Line 498, 501).

This manuscript by Dr. Ding et al., investigated long-term effects of ZIKV-induced maternal immune activation on prenatal and postnatal immunity of offspring, with a focus on neutrophil function. In utero ZIKV exposure was done by injecting a low dose of ZIKV into pregnant C57BL/6 mice at embryonic day 8.5 (E8.5). The authors examined maternal serum cytokine levels, placental immune responses, and fetal development during pregnancy. Notably, male offspring from ZIKV-infected dams showed significantly impaired growth compared to controls. Despite comparable cytokine levels in maternal serum between ZIKV-infected and control groups, ZIKV-induced maternal immunity altered innate immunity in the offspring. Transcriptional profiles of placentas during pregnancy and bone marrow isolated neutrophils, primary innate immune cells, revealed distinctive immune pathways and differential cytokine production profiles in a sex-dependent manner. Different from female offspring, male offspring from ZIKV-exposed dams exhibited a sustained and exaggerated inflammatory response, suggesting impaired regulation of innate immunity. In adult male offspring (at PND35) from ZIKV-exposed dams, male neutrophil response, including ROS production and NETosis, were reduced, while neutrophil response in female offspring remained unaffected. Transcriptional analysis revealed an elevated expression of A20, a negative regulator of NF- κ B signaling and inflammation, in male neutrophils. Studying neutrophils overexpressing A20, the authors discovered that A20 plays as a key sex-dimorphic regulator of neutrophils activation and survival.

The authors concluded that ZIKV triggered maternal immune response has long-lasting effects on the development and innate immunity of offspring. Especially, male offspring exhibit altered inflammatory response and neutrophil dysfunction, which may be attributed to increased A20 expression, a sex-specific regulator of neutrophil activation and function. These findings suggest that ZIKV-induced maternal immunity may increase the vulnerability of male offspring to future infections and inflammatory diseases.

The study employed robust methodologies, including flow cytometry, bulk transcriptomics, followed by RT-qPCR, to analyze innate immune responses in the placenta at E12.5 and E17.5, as well as in bone marrow neutrophils at PND35. The results strongly support sex-dependent alterations in neutrophil activation, recruitment, and function in male offspring following in utero ZIKV exposure. This study further elucidates that the mechanism underlying this sex-dimorphic dysfunction of neutrophils is mediated by A20, a regulator of neutrophil function.

Because the long-term follow-up clinical data from infants and children born from ZIKV-exposed mothers is limited, this study provides important insights into the long-term immune consequences of maternal ZIKV infection on the innate immunity of offspring, particularly regarding neutrophil function in the offspring during pregnancy and the postnatal period.

My comments on the manuscript are the following:

1. Zika viruses do not establish productive infection in B6 mice. Description of the rationale for the use of B6 mice and the dose of ZIKV (50 PFU) is not clearly stated. Justification of intraperitoneal delivery of virus is not described. Authors provide the information in the study design. Has the impact of ZIKV infection with 50 PFU on maternal immunity in pregnant dams been characterized previously? If so, provide reference for the previous publication or provide data as additional information.
2. Some results in Fig2C & D and Fig 3A &B, considering individual variations and heterogeneity among littermate, the sample size, N of 3-5, for transcriptomics is small that may lead to a potentially biased interpretation. Justify the sample size. Are the fetuses from the same single dams or multiple dams? Describe the rationale or limitation of the experimental condition and indicate the number of dams per group in the figure legend.
3. Line 142-144 (Figure 1), it may be worthwhile to elaborate the impact of ZIKV on a sex -dependent fetal growth in mice and consequences of ZIKV infection in the development of children born from ZIKV exposed pregnant women and discuss. Do boys develop more severe developmental defects than girls in humans? Any clinical reports regarding the vulnerability of infants/kids born from ZIKV exposed mothers to COVID-19 or other infections?

Minor comments:

Line 68-71, the current statement “ZIKV exposure *in utero* also have long-term consequences...” to maintain subject-verb agreement, change “exposure” to ‘exposures’ .

Line 268, the statement needs a reference

Line 280, I interpret the results presented in Fig 4 C and D as showing that LPS challenge stimulates neutrophil activation (an increase of CD62L^{lo} cells), which facilitates the egression of neutrophils from bone marrow into bloodstream. Therefore, the levels of mature, activated neutrophils are transiently reduced in the bone marrow and elevated in the blood circulation. If the authors agree, the current description, “... (Fig. 4D), which was consistent with the significant increased neutrophils in the blood stream. ..” may be improved as “... coincided with the significantly increase neutrophils”

Line 410, a typo in ‘time-dependent’

Line 417, a typo 'revealed' instead of "reveled"

Symbols in Fig 6C, Fig 8F and 8G are too small to identify in a print form either use bigger symbols or color coded

Fig8F the y-axis label of CD62L MFI is not legible, log transform the scale would improve readability.